# Light-sheet photonic force optical coherence elastography for high-throughput quantitative 3D micromechanical imaging

Yuechuan Lin [1,2,4], Nichaluk Leartprapun [1,3,4], Justin C. Luo[1,4] & Steven G. Adie [1 ✉]

Quantitative characterisation of micro-scale mechanical properties of the extracellular matrix (ECM) and dynamic cell-ECM interactions can significantly enhance fundamental discoveries and their translational potential in the rapidly growing field of mechanobiology. However, quantitative 3D imaging of ECM mechanics with cellular-scale resolution and dynamic monitoring of cell-mediated changes to pericellular viscoelasticity remain a challenge for existing mechanical characterisation methods. Here, we present light-sheet photonic force optical coherence elastography (LS-pfOCE) to address this need by leveraging a light-sheet for parallelised, non-invasive, and localised mechanical loading. We demonstrate the capabilities of LS-pfOCE by imaging the micromechanical heterogeneity of fibrous collagen matrices and perform live-cell imaging of cell-mediated ECM micromechanical dynamics. By providing access to 4D spatiotemporal variations in the micromechanical properties of 3D biopolymer constructs and engineered cellular systems, LS-pfOCE has the potential to drive new discoveries in mechanobiology and contribute to the development of novel biomechanics-based clinical diagnostics and therapies.

[1] Meinig School of Biomedical Engineering, Cornell University, Ithaca, NY 14853, USA. [2] Present address: Department of Mechanical Engineering, Massachusetts Institute of Technology, Cambridge, MA 02139, USA. [3] Present address: Wellman Center for Photomedicine, Massachusetts General Hospital, Harvard Medical School, Boston, MA 02114, USA. [4] These authors contributed equally: Yuechuan Lin, Nichaluk Leartprapun, Justin C. Luo. ✉email: sga42@cornell.edu

Mechanical properties of the extracellular matrix (ECM) and biological tissues play an important role in regulating cellular functions[1,2]. The ECM not only provides a physical scaffold for cell adhesion and migration[3], but the mechanical properties of the ECM also serve as prominent mechanical cues in various pathophysiological processes[4]. Reciprocally, cells can also alter the micromechanical heterogeneity and stiffness of the ECM via cell-mediated matrix deformation and degradation[5,6]. In the rapidly growing field of mechanobiology, importance has been given to the study of these bi-directional cell-ECM biomechanical interactions in physiologically relevant three-dimensional (3D) environments[7,8]. Given the dynamic nature of biological systems, the ability to characterise 3D spatial *and* temporal variations of ECM mechanical properties at the cellular scale could be useful for understanding the role of biomechanical cell-ECM interactions in physiological processes such as stem cell differentiation[9–11], morphogenesis[12], and wound healing[11], as well as the onset and/or progression of diseases including cancer[13–18], muscular dystrophy[19], and calcific aortic valve diseases[20,21]. Thus, a method for high-throughput 3D quantitative micromechanical imaging of the ECM in engineered cellular systems has the potential to unlock new avenues of research in the field of mechanobiology.

Despite the efforts in the development of a variety of techniques for mechanical characterisation of biological tissues and engineered ECM constructs[22–34], it still remains as a challenge for current techniques to simultaneously support cellular-scale spatially-resolved measurements, practical 3D volumetric acquisition times, and quantitative reconstruction of mechanical properties. Conventional bulk mechanical testing methods such as shear rheometry are unsuitable for spatially-resolved live-cell imaging studies[22]. On the other hand, atomic force microscopy (AFM)—the gold standard for high-resolution mapping of stiffness—is only capable of 2D measurements on the surface of the sample[23]. Emerging optical elastography techniques are under development to fill the gaps between these two extremes. Existing optical coherence elastography (OCE) techniques are most suitable for tissue-level measurements (due to the geometric size of applied mechanical loading) or do not support quantitative reconstruction of viscoelastic properties[24–26]. Brillouin microscopy has demonstrated 3D micromechanical imaging with sub-cellular resolution, but accurate physiological interpretation of the measured GHz longitudinal modulus in relation to the more conventional shear or Young's modulus remains a challenge[27–29]. Particle-tracking passive microrheology is capable of quantitative measurements in 3D, but it is predominantly applicable in highly compliant (shear modulus <100 Pa) viscosity-dominant materials[30–34]. Alternatively, optical tweezer-based active microrheology (OT-AMR) can support microrheological measurements in more rigid elasticity-dominant viscoelastic materials[30–34]. However, the practicality of OT-AMR for volumetric measurements is still limited due to the need to serially align (with 0.1-µm precision) the high numerical aperture (NA) trapping and detection beams to individual probe beads that are randomly distributed in 3D space.

We present light-sheet photonic force OCE (LS-pfOCE), an all-optical method based on a novel use of photonic radiation pressure from a light sheet for alignment-free parallel optical micro-manipulation, to address the persisting need for high-throughput quantitative 3D micromechanical characterisation in the field of mechanobiology. LS-pfOCE can support volumetric imaging with quantitative reconstruction of viscoelasticity in 3D physiologically relevant ECM constructs and engineered cellular systems. We validate and demonstrate LS-pfOCE in polyacrylamide (PAAm) gels, collagen matrices with varying fibre architecture, and live-cell imaging of cell-mediated ECM remodelling in 3D fibrin hydrogels seeded with NIH-3T3 fibroblast cells.

## Results

**Principle of photonic force OCE.** Inspired by the pioneering work of Nobel laureate Arthur Ashkin[35], we exploited a less conventional mode of optical manipulation, based on radiation pressure from a low-NA beam, to develop photonic force (PF)-OCE for 3D mechanical microscopy[36]. PF-OCE utilises radiation pressure from a low-NA beam (NA ≤ 0.4, compared to NA ≥ 1 in conventional high-NA OTs) to provide "AFM-like" localised mechanical loading that is applied to micron-sized probe beads randomly distributed in 3D space. Leveraging interferometric detection from phase-sensitive optical coherence tomography (OCT), oscillations of these probe beads induced by harmonically modulated radiation pressure can be detected with sub-nanometre displacement sensitivity after compensating for the confounding photothermal (PT) response of the medium. The measured complex mechanical responses of individual probe beads reflect the local viscoelastic properties of the medium in the vicinity of each bead (i.e., bead "microenvironment"). With the extended depth coverage afforded by the use of a low-NA beam, and no prerequisite for precise bead-wise optical alignment that restricts the practicality of OT-AMR, PF-OCE is well-suited for volumetric measurements in 3D engineered cell culture systems, with additional benefits of OCT providing label-free, rapid volumetric imaging in scattering media. Nevertheless, our previous implementation of PF-OCE for 3D mechanical microscopy, based on transverse raster-scanning of a Gaussian forcing beam, resulted in a pulse-train temporal excitation profile on each probe bead instead of the desired continuous sinusoidal waveform (see Supplementary Fig. 2b in ref. [36].). The pulse-train excitation, with the duty cycle determined by the "dwell time" of the excitation beam on each bead, is: (1) inefficient due to over an order of magnitude reduction in time-averaged force exerted on each probe bead compared to continuous excitation[36], and (2) does not readily support microrheological quantification of viscoelasticity[37,38] due to the presence of higher harmonics in the comb-like excitation profile. Thus, a more efficient localised mechanical excitation scheme is needed in order to realise practical volumetric quantitative micromechanical imaging with PF-OCE. Inspired by the transformative impact of light-sheet microscopy for high-throughput bio-microscopy, we leverage photonic radiation pressure from a light sheet, as opposed to a standard focused Gaussian beam, to achieve efficient parallel mechanical excitation. LS-pfOCE addresses the critical limitations of the original Gaussian-beam PF-OCE and dramatically improves the capability for quantitative volumetric time-lapsed micromechanical imaging.

**Characterisation and validation of the LS-pfOCE system.** The LS-pfOCE system adopts a pump-probe configuration. A weakly focused light sheet generated by a cylindrical lens is used for radiation-pressure excitation, instead of a Gaussian beam (Fig. 1a, see Supplementary Fig. 1 for a detailed system schematic). The light sheet has a measured full-width half-maximum (FWHM) dimension of 80 µm × 1.4 µm at the focal plane (Fig. 1b) and a total beam power of 120 mW at the sample surface, providing a light sheet radiation-pressure force with a peak of ~3 pN and a FWHM spatial extent of 80 µm in the lateral (long-axis) and axial dimensions (Fig. 1c, d). (The deviation between theoretical simulation and measured force profiles in Fig. 1c is attributed to non-idealities in the optical system such as optical aberrations[39].) By harmonically modulating its power, the light sheet PF beam can exert continuous sinusoidal radiation-pressure force on multiple probe beads located within the span of its long axis in parallel, without the need for beam-scanning (Fig. 1a, inset). The

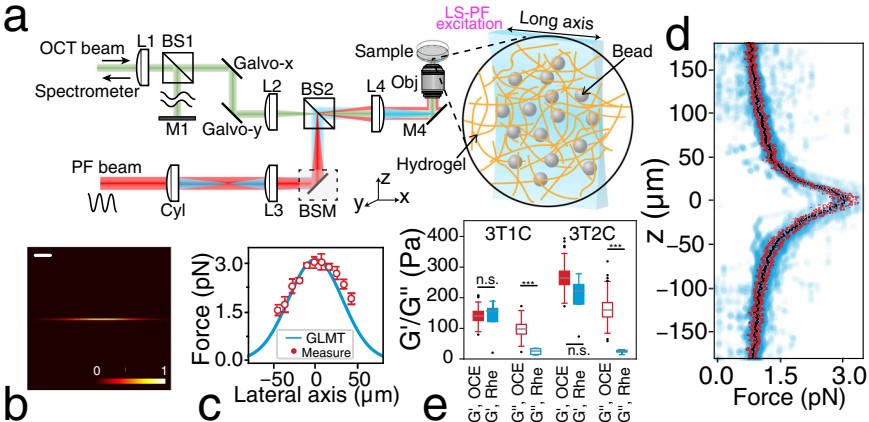

**Fig. 1 LS-pfOCE system characterisation and validation. a** Schematic of the LS-pfOCE system. Light-sheet radiation-pressure excitation is generated by a cylindrical lens and combined in free-space with the sample arm of a spectral-domain OCT system for interferometric detection of radiation-pressure-induced bead oscillations. See "Methods" for a full description of the LS-pfOCE system. L: lens, Cyl: cylindrical lens, BS: beam splitter, BSM: beam-steering module, M: mirror, Obj: microscope objective. **b** Normalised light-sheet intensity profile measured at the focal plane. Scale bar = 20 μm. **c** Measured (red, mean ± standard deviation from $N = 48$ beads) and simulated (blue, Generalized Lorenz-Mie Theory (GLMT), see Supplementary Method 6) lateral light-sheet radiation-pressure profile at its focal plane on 1.9-μm melamine-resin beads. **d** Measured axial light-sheet radiation-pressure profile at its lateral centre on 1.9-μm melamine-resin beads, showing data (blue) and depth-dependent median (black) and median absolute difference (red) from $N = 48$ beads. See "Methods" for light-sheet radiation-pressure force measurement procedure. **e** Comparison of $G'$ and $G''$ in PAAm gels measured by 3D LS-pfOCE (red, OCE) and bulk shear rheometry (blue, Rhe) at 20 Hz. Each box represents data from $N = 5$ samples for rheometry and $N = 225$ and 263 beads for LS-pfOCE in 3T1C and 3T2C gels, respectively. Horizontal line indicates median, box spans 1st to 3rd quartile, and whisker length corresponds to 1.5 times the interquartile range. Significant difference in group means (two-sided Welch's t-test) is indicated by \*\*\*$p$ value < 0.001 and n.s. $p$ value > 0.05.

radiation-pressure-induced bead oscillations are detected by a phase-sensitive OCT, whose fast-axis scanning is parallel to and co-aligned with the long axis of the light sheet, operating in a 2D BM-mode acquisition scheme at each slow-axis position (see "Methods" for specific acquisition parameters). Volumetric imaging was accomplished by translating the sample with a motorized actuator stage in a direction perpendicular to the long axis of the light sheet (equivalent to the slow-axis scanning of the OCT system). Under this acquisition scheme, the light-sheet implementation provides a 7× improvement in the time-averaged radiation-pressure force exerted on each probe bead over the equivalent Gaussian-beam implementation with the same PF beam power. The presented LS-pfOCE acquisition scheme supports quantitative 3D micromechanical imaging up to a maximum shear elastic modulus of 550–1200 Pa (with an experimental displacement sensitivity of 36–76 pm for OCT signal-to-noise ratio ≥ 28 dB) over a volumetric field-of-view (FOV) of 80 μm × 350 μm × 80 μm (fast axis × slow axis × depth), which typically contains several hundreds of embedded probe beads (the *en face* FOV can be readily adjusted by extending the range of the slow-axis scan; see "Methods" and "Discussion").

Light-sheet radiation-pressure-induced complex mechanical responses of individual probe beads were measured via phase-sensitive OCT following compensation for the PT response of the medium. Compared to our previous work[36], in order to improve the compatibility for live-cell imaging applications, LS-pfOCE implements a more robust PT response compensation approach that eliminates the need to incorporate exogenous PT reporters into the sample[36]. The micromechanical properties (i.e., the local complex shear modulus $G^* = G' + iG'$ in the "microenvironment" of each bead) can be reconstructed from the measured bead-wise complex mechanical response and the experimentally measured light-sheet radiation-pressure force profiles (Fig. 1c, d; Supplementary Method 2 provides a detailed description of the LS-pfOCE reconstruction procedure). Quantification of $G^*$ by LS-pfOCE is validated in homogeneous PAAm gels (see "Methods" for sample preparation) by comparison to parallel-plate shear rheometry (Fig. 1e). Under the premise that the mesh size of the PAAm polymer network is smaller than the size of LS-pfOCE probe beads (1.7-μm diameter), a micro-to-macro-scale comparison can reasonably be made between the storage modulus, $G'$, measured by LS-pfOCE and shear rheometry[37]. The measurement of $G'$ by LS-pfOCE is in good agreement with that of shear rheometry at the same modulation frequency of 20 Hz. However, LS-pfOCE measurement of the loss modulus, $G''$, differs from that of shear rheometry due to distinct viscous responses (e.g., viscous drag of fluid flow through the porous polymer network) at the micro- versus macro-scale, which is in agreement with our prior PF-OCE work based on a Gaussian-beam excitation[37,38] as well as other micro-scale methods[40].

**Imaging the micromechanical heterogeneity of fibrous ECM.** We demonstrate quantitative characterisation of micromechanical heterogeneity in ECM constructs by performing LS-pfOCE in three microscopically heterogeneous fibrous collagen matrices with different microarchitectures (see "Methods" for sample preparation) and homogeneous PAAm hydrogel. LS-pfOCE is able to reveal the micromechanical heterogeneity of the fibrous collagen matrix compared to the homogeneous PAAm gel (Fig. 2a), where the distributions of both $G'$ and $G''$ measured within a sample are broader (i.e., larger FWHM) for collagen than PAAm (Fig. 2b). LS-pfOCE measurements of $G'$ (elasticity or "stiffness") and $R = G''/G'$ (relative viscosity or "loss ratio") are also able to reveal the distinct microarchitectural characteristics of the three collagen matrices (Fig. 2c). The fibre microstructure in each sample was statistically analysed using CT-FIRE software[41] and the high-resolution confocal reflectance images (top row in Fig. 2c), and shown in Supplementary Data 5. The most microarchitecturally homogeneous collagen sample C1, with the thinnest and shortest collagen fibres, exhibits an overall lower stiffness and little variability in both $G'$ and $R$. In contrast, samples C2 and C3, with thicker and longer collagen fibres, exhibit an overall higher stiffness with significant variability in both $G'$ and $R$. Quantitatively, samples C2 and C3, with the more heterogeneous

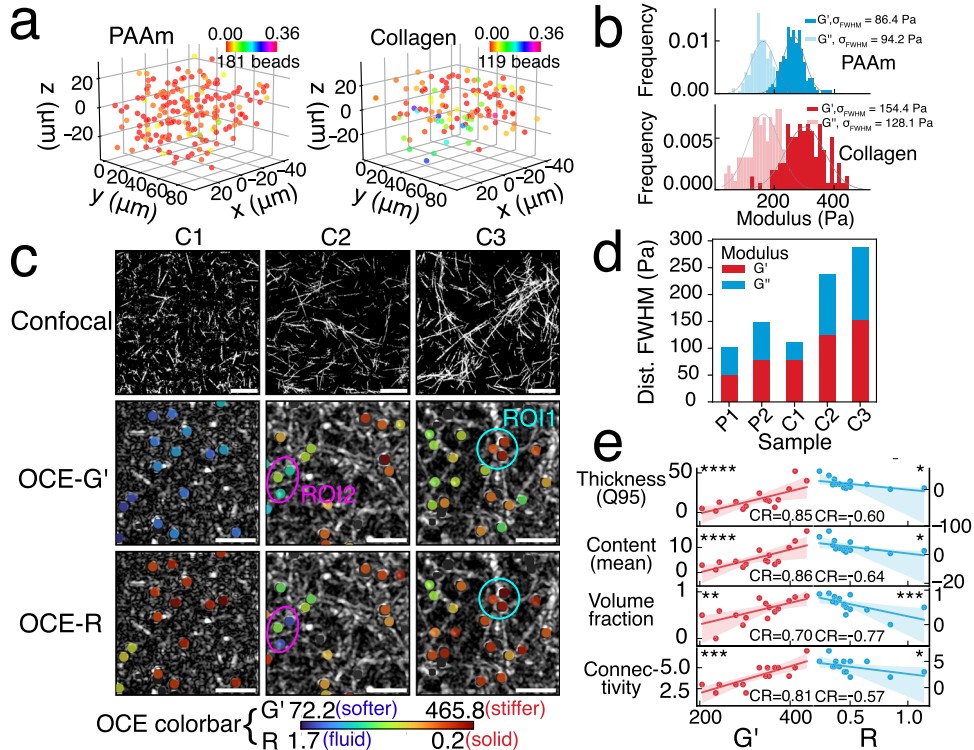

**Fig. 2 Characterisation of micromechanical heterogeneity in fibrous collagen matrices. a** Normalised relative variation in stiffness (defined as $[(G' - G'_{median})/G'_{median}]^2$) measured by 3D LS-pfOCE in a PAAm gel (3T2C, left) and collagen matrix (C3, right), where larger variation is observed in collagen. $N = 181$ and 119 beads for PAAm and collagen, respectively. **b** Histograms and statistical distributions of absolute (i.e., without normalisation) $G'$ and $G''$ measurements in **a**. $\sigma_{FWHM}$ indicates the FWHM of the distributions. *En face* confocal reflectance microscopy (top, FOV 106 μm × 106 μm) and resolution-enhanced OCT images (following procedure in ref. [68]) overlaid with colour-coded LS-pfOCE measurements of $G'$ (middle) and $R$ (bottom) at the focal plane. Confocal images serve as higher-resolution references for visualising the microstructures of each collagen sample and are not co-registered with OCE images. ROI1 (cyan circles) indicates a stiffer and more solid-like microenvironment in the presence of thick collagen fibres. ROI2 indicates a more compliant and fluid-like microenvironment in the absence of any clear collagen fibres. All OCE images show a FOV of 71 μm × 71 μm. Scale bar, 20 μm. **d**, FWHM of the distributions of LS-pfOCE $G'$ (red) and $G'$ (blue) in PAAm gels and collagen matrices. $N = 137$, 107, 105, 132 and 100 beads for P1 (3T1C), P2 (3T2C), C1, C2, and C3, respectively. **e** Correlation of LS-pfOCE $G'$ (red) and $R$ (blue) to measures of collagen matrix microarchitectural characteristics. $N = 15$ beads in sample C3. Q95 and Mean indicate the 0.95 quantile and mean of OCT scattering intensity within a 3-μm radial distance from the circumference of each bead, respectively. Fibre volume fraction likewise represents fraction within the 3-μm radius around each bead. Fibre connectivity represents the number connected fibre branches at each "node" (i.e., bead) (see "Methods" for further details). CR indicates the Spearman rank correlation coefficient; significance of the correlation is indicated by *$p$ value < 0.05, **$p$ value < 0.01, ***$p$ value < 0.001 and ****$p$ value < 0.0001. Solid line and shaded region represent the linear best-fit line and 95% confidence interval, respectively.

microarchitecture, have a broader distribution for both $G'$ and $G''$ compared to that of sample C1 and the PAAm gels (Fig. 2d).

The ability to microrheologically quantify viscoelasticity (i.e., both $G'$ and $G''$) of LS-pfOCE also reveals an interesting observation: the variations in $G'$ and $R$ tend to follow opposite trends in the more fibrous C2 and C3 samples. For instance, region of interest (ROI) 1 (cyan circles in collagen sample C3 of Fig. 2c), where the microenvironment is dominated by a junction of thick collagen fibres with multiple connected branches, exhibits higher stiffness (higher $G'$) and a more "solid-like" behaviour (lower $R$). On the other hand, ROI2 (magenta circles in collagen sample C2 of Fig. 2c), where the microenvironment is absent of any clearly resolvable collagen fibres, exhibits lower stiffness (lower $G'$) and a more "fluid-like" behaviour (higher $R$). This observation may be supported by the biphasic description of a porous hydrogel, where the collagen fibres form the solid phase with an elastic response (which dominates in ROI1) and the rest of the material form the fluid phase with a viscous response (which dominates in ROI2). These results may also support the network connectivity interpretation[42–44] and rigidity percolation behaviour[45,46] of fibre network models—ROI1 contains higher volume fraction of collagen with 3–6 connected collagen branches

(thus, exhibiting higher node stiffness), whereas ROI2 contains lower collagen volume fraction with 0–2 connected branches at most (thus, exhibiting lower node stiffness). To corroborate these interpretations, LS-pfOCE measurement in sample C3 is correlated to four metrics describing the microarchitectural characteristics of the microenvironment surrounding each bead (Fig. 2e, see "Methods" for details on the calculation of each metric). Collagen fibre thickness and overall collagen content were inferred from the 0.95 quantile (Q95) and mean of OCT scattering intensity, respectively. $G'$ and $R$ show a strong (correlation coefficient > 0.6) and statistically significant correlation to both metrics. Consistent with existing mathematical models of fibrous biopolymer[42–44], $G'$ increases with increasing local fibre volume fraction and fibre network connectivity. However, whereas these mathematical models only consider elasticity, our results also show a negative correlation of these network parameters with $R$. Quite remarkably, while fibre thickness, content, and connectivity are more strongly correlated (higher CR) to $G'$ than $R$, the opposite is true for fibre volume fraction—that is, compared to elasticity, the viscous response of the network *at the micro-scale* is more strongly influenced by the "absence" of fluid. Overall, the results in Fig. 2 demonstrate the

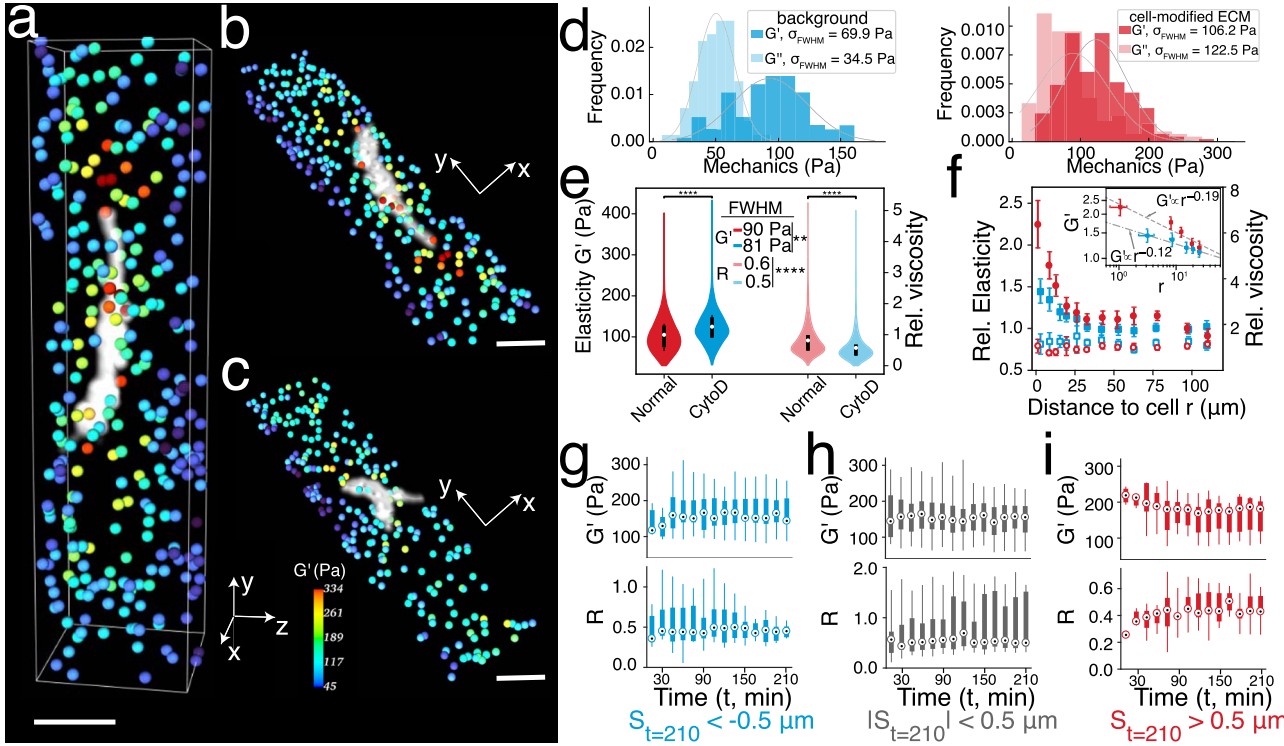

**Fig. 3 Live-cell imaging of cell-mediated spatiotemporal variations in ECM micromechanical properties. a** 3D LS-pfOCE measurements of $G'$ (colour-coded beads) around a cell (white) and $z$-projection of 3D measurements under **b**, normal condition and **c**, 2 h after treatment with CytoD (**b** and **c** show two different cells). Size of the beads is not to scale with the cell body. (DMSO control of CytoD treatment is provided in Supplementary Data 1. The normal condition in **b** did not receive DMSO, see "Methods" for sample preparation and live-cell imaging protocol.) All images share the same colour scale. (See Supplementary Videos 1 and 2 for visualisation of **b** and **c** from different viewing angles). Scale bar, 50 μm. **d** Histograms and distribution fits of 3D LS-pfOCE measurements in the pericellular (red, $N = 205$ beads) and cell-free regions (blue, $N = 97$ beads) from 1 cell. $\sigma_{FWHM}$ indicates the FWHM of the distributions. **e** Violin plots of 3D LS-pfOCE measurements around cells under normal (red, $N = 1824$ beads) and CytoD (blue, $N = 1567$ beads) conditions from 8 cells. Marker indicates median and black bar spans 1st to 3rd quartile. The FWHM of each distribution is stated. Significant difference in group means (two-sided Welch's $t$-test) and variance (Levene's test) is indicated by **$p$ value < 0.01 and ****$p$ value < 0.0001. **f** 3D LS-pfOCE measurements of $G'$ (solid marker) and $R$ (open marker) from the data in **e**, normalised by the measurements in faraway background regions ($\geq 30$ μm from cell) in each sample, as a function of distance to the cell under normal (red) and CytoD (blue) conditions. Inset shows power law fits within <30 μm to cell; the decay exponent has a 95% confidence interval of –0.19 ± 0.08 (red) and –0.12 ± 0.003 (blue). Each data point represents mean ± standard deviation from $N \geq 20$ beads. **g–i** Boxplots of $G'$ (top) and $R$ (bottom) from 5 cells as a function of time after CytoD treatment in regions that experience high-negative (blue, $N = 10$ beads), low (grey, $N = 16$ beads), and high-positive (red, $N = 8$ beads) matrix deformation $S$, respectively. Marker indicates median, box spans 1st to 3rd quartile, and whisker spans the full range of values.

capability of LS-pfOCE for quantitative micromechanical characterisation of microscopically heterogeneous fibrous ECM constructs, which can enable correlative analysis between micromechanical and microarchitectural heterogeneities in the study of biopolymer mechanics.

**Imaging the cell-mediated ECM micromechanical remodelling.** We demonstrate potential mechanobiological applications of LS-pfOCE for live-cell imaging studies of cell-mediated ECM remodelling in 3D fibrin constructs seeded with NIH-3T3 fibroblasts (see "Methods" for sample preparation). LS-pfOCE is able to quantitatively characterise, in 3D, the spatial variations in the micromechanical properties of the fibrin ECM surrounding isolated cells (Fig. 3a, b). Higher $G'$ can be observed closer to the cell body (especially extending from the tip of the cell) compared to regions further away, which corroborate with previous results from OT-AMR[32,33]. Quantitatively, both the mean $G'$ value and the width of the distributions of $G'$ and $G''$ measurements are larger in the cell-modified ECM (i.e., pericellular space) than the native (i.e., cell-free region) fibrin ECM (Fig. 3d). LS-pfOCE is also able to characterise changes in the pericellular micromechanical properties due to altered cellular activity

(the presence of cells by itself (i.e., as "objects") is not expected to alter ECM micromechanics[37]). As an example, we investigated the consequences of reduced cellular contractility via treatment with Cytochalasin D (CytoD), an inhibitor of actin polymerisation (Fig. 3c, measurement taken 2 h after CytoD treatment, see "Methods" for the live-cell imaging protocol). Compared to the normal condition (Fig. 3b), the CytoD-treated sample exhibits strongly reduced stiffness in the pericellular space (especially at the cell tip) accompanied by a slight increase in stiffness further away from the cell (see Supplementary Data 6 for quantitative comparison of the background regions), resulting in a relatively uniform spatial distribution of $G'$ (Fig. 3c). Thus, significant differences are observed in both the group means and variances of $G'$ and $G''$ measurements between the normal and CytoD conditions (Fig. 3e). Control experiments with pure DMSO (solvent/carrier for CytoD treatment) and measurements in blank fibrin constructs (not seeded with cells) are provided in Supplementary Data 1 (see experimental details in "Methods").

LS-pfOCE is also able to quantitatively characterise the variations in $G'$ and $R$ as a function of distance to the cell body in 3D space (see "Methods" for details of this analysis). A decreasing trend in $G'$ as a function of distance is apparent within

the pericellular space (within 30 μm of cell the boundary) in both normal and CytoD conditions (Fig. 3f, solid markers; and Supplementary Data 2). However, the stiffness gradient is significantly steeper in the normal condition (decay exponent of −0.19 for normal versus −0.12 for CytoD condition, see inset of Fig. 3f), suggesting that the degree of cell-mediated pericellular stiffening is reduced (as expected) after the inhibition of actin polymerisation. The weaker stiffness gradient that remains after CytoD treatment may be attributed to the accumulated increase in matrix density in the pericellular space (i.e., cell-mediated accumulation of fibrin during the 12-h incubation period under normal conditions, prior to the CytoD treatment)[33]. Compared to $G'$, the variation in $R$ as a function of distance to cell is less apparent and does not follow an obvious monotonic trend (Fig. 3f, open markers). However, a mild increasing trend can be observed within the pericellular space (≤30 μm to cell), opposing the decreasing trend in $G'$, in the normal condition.

The contributions of cell-force-mediated nonlinear ECM stiffening (i.e., strain- and/or stress-stiffening response of the fibrous ECM induced by cellular contractility) versus fibrous concentration-dependent ECM stiffening (i.e., accumulation of higher matrix density around cell) to the observed pericellular stiffness gradient have been speculated in a previous OT-AMR study, where similar results were observed after CytoD treatment[33]. Here, our 3D LS-pfOCE results also reveal a small increase in $G'$ after the CytoD treatment in the intermediate ECM (≥30 μm away from the cell boundary, see Supplementary Data 3) —a small difference which could not be clearly discerned from previous bead-wise OT-AMR measurements[33]. We hypothesise that this mild stiffening in the intermediate ECM is related to the redistribution of the fibrin matrix—from where the matrix is concentrated by the contractile cell in the pericellular space (≤30 μm to cell), back into the intermediate regions (≥30 μm to cell)—following the inhibition of cellular contractility. Compared to the OT-AMR study[33], LS-pfOCE facilitates a more complete, high-throughput volumetric characterisation of cell-mediated spatial variations in the micromechanical properties of surrounding ECM by providing quantitative measurements of both elasticity and viscosity in 3D (as opposed to serially measuring a selected number of probe beads, typically within a 2D plane).

LS-pfOCE also enables time-lapse monitoring of the spatio-temporal dynamics of micromechanical properties in live cellular systems. As an example, we investigated the temporal variations in the pericellular micromechanical properties (near the tip of the cell) as the CytoD treatment takes effect (see "Methods" for time-lapse imaging protocol). Time-lapse LS-pfOCE is able to simultaneously track the changes in both micromechanical properties and local matrix deformations (i.e., displacements of probe beads, see "Methods" for details of this calculation), which reveals statistically significant correlation (Supplementary Data 4). This result is consistent with the hypothesis that the effect of CytoD treatment on pericellular micromechanical properties is governed by the spatiotemporal dynamics of the inhibition of actin polymerisation, which in turn impacts cell-force-mediated matrix deformations.

Our time-lapse LS-pfOCE measurements also reveal an interesting behaviour that, to our knowledge, has never been reported by other techniques. While previous studies in the field have only reported the softening of the ECM following contractility inhibition[32,33], time-lapse LS-pfOCE shows that the changes in micromechanical properties and local matrix deformation occur in both positive *and* negative directions. Specifically, regions of the matrix that experience a negative deformation tend to exhibit an initially lower $G'$ (~100 Pa, similar to the native fibrin in the cell-free regions in Fig. 3d) that increases over time (Fig. 3g). Conversely, regions of the matrix

that experience a positive deformation tend to exhibit an initially higher $G'$ (>200 Pa, similar to the cell-modified pericellular fibrin in Fig. 3a, b) that decreases over time; this is also accompanied by an opposite trend in $R$ (Fig. 3i). The latter behaviour is consistent with previous reports[32,33], and may be attributed to the "relaxation" of cell-force-mediated ECM stiffening governed by CytoD inhibition of actin polymerisation[47]. Notably, both $G'$ and $R$ stabilise after approximately 2 h, consistent with the amount of time taken for the total cell force to approach zero following CytoD treatment in a previous traction force microscopy study[47]. The former behaviour may be attributed to the redistribution of fibrin network in the pericellular space (see discussion of Fig. 3f above) as well as the cell-mediated ECM stiffening by any residual actin filaments that have yet to be affected by CytoD (noting that attachment proteins on the cell surface to the ECM can remain even after the inhibition of actin polymerisation). Although this behaviour has not been reported previously, it could have important implications in some mechanobiological studies[48,49].

Meanwhile, regions of the matrix that experience minimal deformation in either direction also tend to exhibit smaller changes in $G'$ and an overall higher $R$ (Fig. 3h), which suggests that the microarchitecture of these regions may be dominated by large fluid-filled pores. Remarkably, all three types of dynamic responses stabilise to a similar level ($G'$ ~160 Pa and $R$ ~0.4) over time after CytoD inhibition of cellular contractility, even though they initially exhibit distinct cell-mediated ECM micromechanical properties (Fig. 3g–i). Overall, the results in Fig. 3 demonstrate that the live-cell imaging capability of LS-pfOCE can provide access to previously unavailable measurements, which emphasises its potential to support new discoveries in the field of mechanobiology.

## Discussion

LS-pfOCE harnesses photonic radiation pressure from a light sheet, and leverages label-free tomographic imaging via phase-sensitive OCT, to perform high-throughput quantitative 3D micromechanical imaging in biological hydrogels and live cellular systems. The unprecedented combination of light-sheet optical manipulation and state-of-the-art interferometric displacement measurements provides a foundation for future studies and development of photonic radiation pressure, including opening a new application area of light-sheet optics in elastography/microrheology. Compared to our previous work[37,38], the dramatically improved imaging capabilities of LS-pfOCE rely on two key innovations—a parallel radiation-pressure force scheme that enables volumetric imaging with continuous harmonic excitation on each individual probe bead, and a more robust PT compensation approach that eliminates the need for exogenous PT reporters (see "Methods").

To exemplify the potential utility of LS-pfOCE in the broad fields of mechanobiology and biophysics, we demonstrate its unique micromechanical imaging capabilities in well-established ubiquitous experimental systems in these fields. We applied LS-pfOCE to characterise the micromechanical heterogeneity in fibrous collagen matrices and—for the first time—characterised the 3D spatial variations and temporal dynamics of cell-mediated ECM micromechanical properties via live-cell imaging of fibroblasts within 3D fibrin constructs. The 3D imaging of cell boundaries based on OCT speckle fluctuations[8,47,50,51], as well as ECM deformations and micromechanical properties allows the measurements of pericellular mechanical properties to be interpreted within the context of the cell orientation (i.e., extension direction) and cell-induced ECM deformations. Importantly, even in "familiar" settings, LS-pfOCE was able to support new analysis and glean information that has never been reported by existing

techniques in the same experimental systems (e.g., the correlation between collagen micromechanics and microarchitecture in Fig. 2e, and measurements of time-dependent stiffening and softening of pericellular viscoelasticity following Cytochalasin D treatment that are correlated to cell-induced matrix deformation in Fig. 3g–i). The results presented here form the basis for future applications of LS-pfOCE as a potentially powerful tool to investigate profound mechanistic phenomena in mechanobiology and soft materials.

Although the fast-axis scan range and depth coverage of the current LS-pfOCE system is limited by the lateral extent (long-axis width) and the NA (short-axis focusing) of the light sheet, a larger volumetric FOV can be achieved by simply tuning the long-axis width of light-sheet PF beam. More sophisticated light-sheet generation optics can be introduced to generate laterally uniform light-sheet PF excitation with an extended depth of focus, such as those previously developed for light-sheet fluorescence microscopy[52–54]. A laterally uniform light-sheet radiation-pressure force profile will be advantageous because it can support the same maximum measurable stiffness over the entire FOV. However, this is not a prerequisite for LS-pfOCE because our 2D force measurement method (see "Methods") allows a non-uniform force profile to be accounted for in the quantitative reconstruction of complex shear modulus. In addition, the volumetric throughput of LS-pfOCE can be further improved by increasing the mechanical response amplitude-to-noise ratio[37,38] (such as by increasing the power of the PF beam or reducing the OCT system noise), which would allow reliable measurements of bead mechanical response to be made with a lower number of temporal frames per slow-axis position[37,38]. One way to reduce noise in the current LS-pfOCE system is by replacing the motorised actuator with a high-fidelity stepper motor. Additional improvements may be realised by implementing common-path phase-sensitive detection[55].

The acquisition scheme of LS-pfOCE can be flexibly tailored to best fit the experimental needs of specific mechanobiological applications—by adjusting the number of temporal (BM-mode) frames acquired at each slow-axis position. This navigates the trade-off between acquisition time and the maximum stiffness that can be reliably quantified (i.e., the displacement sensitivity supported by a given number of temporal frames[37,38]). Meanwhile, LS-pfOCE may also be implemented with other high-speed OCT imaging approaches such as swept-source and parallel line-field OCT in order to further improve its imaging speed[56]. A faster option for OCT beam-scanning, such as a resonant scanner for multi-MHz-OCT[8,47,50,51] or even a MEMS scanner[56] (since LS-pfOCE does not require a large scanning angle to span the long axis of the light sheet), that can support BM-mode frame rate in the kHz range will significantly improve the acquisition speed overall (see Supplementary Discussion 1 for further considerations related to high-speed imaging).

On the one hand, from an imaging standpoint, the spatial resolution of LS-pfOCE is dictated by statistical sampling of the randomly distributed probe beads[36]. Here, we employed average edge-to-edge spacing of 12–14 μm, with the minimum spacing being limited by the allowable bead volume fraction that would not interfere with cellular behaviour. A smaller bead size would hypothetically allow for denser bead spacing (thus, finer spatial sampling), but at the cost of lower light-sheet radiation-pressure force magnitude (thus, lower maximum measurable stiffness). Therefore, in addition to the acquisition parameters discussed above, considerations of bead size and bead spacing should also be made in order to optimize LS-pfOCE for each specific application. On the other hand, from a mechanical characterisation standpoint, $G^*$ represents the average mechanical properties within the microenvironment surrounding each bead, extending as far as the mechanical interaction length[57] of the induced bead oscillation. Given the micrometre-scaled bead size (i.e., highly localised mechanical excitation[57]) and the sub-nanometre bead oscillation amplitude in our experiments, this spatial extent is expected to be much smaller than the bead spacing. That is, the bead oscillation-induced deformation of the surrounding medium would rapidly drop below the displacement noise floor away from the bead, effectively isolating our measurements to the immediate vicinity of each bead. At both limits (average bead spacing, and mechanical interaction length), LS-pfOCE provides an unprecedented spatial resolution compared to existing 3D quantitative mechanical characterisation approaches[24–26].

LS-pfOCE has the potential to enable numerous novel research studies in the field of mechanobiology. Recent developments in traction force microscopy (TFM) have enabled the study of time-varying cell forces and cell-mediated ECM deformations in 3D, including both isolated[8,47,50,51] and collective cellular behaviours (such as stromal-cell-mediated dissemination of cancer cells from co-cultured tumour spheroids[6,58,59] and epithelial-mesenchymal transitions in multicellular epithelial clusters[60]). LS-pfOCE has the potential to significantly elevate such studies by providing crucial (albeit currently missing) information on the 4D spatio-temporal dynamics of the ECM micromechanical properties to complement TFM-based measurements. Our correlative analysis of the changes in ECM viscoelasticity and matrix deformation in Fig. 3g–i and Supplementary Fig. 12 is an example of how new mechanobiological insights may be obtained with the combination of LS-pfOCE and TFM-based measurements. At the more fundamental level, LS-pfOCE also has the potential to enable a significantly more accurate force reconstruction in TFM (which typically assumes that the ECM is linearly elastic, static, and homogeneous[61]) by providing 4D spatiotemporal measurements of ECM viscoelasticity as an input. Furthermore, LS-pfOCE can be integrated into a multimodal microscopy platform that incorporates confocal or light-sheet fluorescence microscopy to enable simultaneous characterisation of micromechanics and molecular signalling pathways.

Moreover, ongoing efforts in the field of biophysics are focused on the development of computational models to describe and unravel the complexities of biopolymer mechanics, including fibrous ECM constructs[5,62–66]. LS-pfOCE has the potential to help inform and validate the development of these computational models by providing experimental quantification of the micromechanical properties of biopolymer constructs. Our correlative analysis of collagen micromechanical properties and microarchitecture in Fig. 2e provides a preliminary example of new information accessible through the combination of microstructural OCT imaging and LS-pfOCE that may be useful for future research on biopolymer mechanics. To this end, future applications of LS-pfOCE may extend beyond the scope of linear viscoelasticity, to also investigate poroelasticity and anisotropy (e.g., by rotating the sample to exert force from different directions) of fibrous biopolymer networks.

## Methods

**LS-pfOCE system and beam alignment procedure**. The LS-pfOCE system consisted of a spectral-domain (SD)-OCT imaging system and a light sheet for radiation-pressure force excitation, which is combined into the OCT sample arm in a pump-probe configuration (see Supplementary Method 1 for detailed system schematic). The OCT system was sourced by a broadband superluminescent diode (Thorlabs, LS2000B) with a centre wavelength and a FWHM bandwidth of 1300 nm and 200 nm, respectively. Spectral data was detected by a spectrometer (Wasatch Photonics, Cobra 1300) with a bandwidth of 245 nm and a 2048-pixel line-scan camera (Sensors Unlimited, GL2048). The sample arm utilised a double-pass illumination/collection configuration with an inverted 20× microscope objective (Olympus, LCPLN20XIR) with an NA of 0.45. Telecentric beam-scanning was accomplished with a 2-axis galvanometer (Cambridge Technology, ProSeries 1, 10 mm) and a unit-magnification telescope, which imaged the galvanometer to the

back focal plane of the objective. The transverse and axial resolutions (in air) of the OCT system were 2.3 μm and 3.4 μm, respectively. The light-sheet pump beam was generated by a laser diode (Frankfurt Laser Company, FLU0786M250, HI780 fibre output) at a wavelength of 789 nm and a cylindrical lens (Thorlabs, ACY254-100-B). The light sheet was injected into the telescope of the OCT system via a longpass dichroic mirror (Thorlabs, DMLP1180R) and shared the same objective. The long axis of the light sheet was collimated with a FWHM beam width of 80 μm while the short axis was focused to a FWHM beam width of 1.4 μm. The power of the light-sheet pump beam was harmonically modulated with a peak power of 120 mW measured directly after the objective. 3D imaging was accomplished via galvanometer scanning of the OCT beam along the fast axis and a motorised actuator (Thorlabs, ZST225B motor, KST101 controller, and KCH601 controller hub/power supply) stepping the sample along the slow axis, while the light-sheet pump beam remained stationary.

A beam-steering module ensured co-alignment between the pump and probe beams such that (1) the optical axes of both beams were parallel and overlapped, (2) the focal planes of both beams were ≤10 μm from each other inside the sample, and (3) the long axis of the light sheet was parallel to the fast galvanometer scanning axis of the OCT system. The beam alignment was visualised in real-time by placing a CCD camera (Thorlabs, DCC1545M) above the objective to image the light sheet and OCT beam spots at different axial position relative to the focal plane. After real-time beam alignment, the light-sheet radiation-pressure force was measured as a final LS-pfOCE system performance check prior to each experiment.

**LS-pfOCE radiation-pressure force measurement**. The measurement of radiation-pressure force from the light-sheet adapts an OCT-based depth-resolved force measurement method[39], with modifications to extend the previous 1D measurement to 2D in order to characterise both the axial and lateral (along the light-sheet long axis) force profiles (Fig. 1b, d). A 10% w/w solution of glycerol and water (refractive index 1.3469, close to those of hydrogels, see Table 1 in ref. [39]. for other relevant physical properties) was used as the viscous fluid medium for the force measurements. The 2D measurement was accomplished via a BM-mode acquisition scheme (instead of M-mode in ref. [39].), with the fast-axis scanning parallel to the long axis of the light sheet. The "M-mode-equivalent" space-time axial bead trajectory images were obtained across the BM-mode frames (i.e., temporal dimension) at different lateral segments along the fast axis (i.e., long axis of the light sheet). The depth-resolved axial radiation-pressure force profile at each lateral segment was reconstructed by tracking the instantaneous axial velocity and acceleration of each bead from the space-time axial bead trajectory image, then, solving the 1D equation of motion along the axial direction using known fluid viscosity and other physical properties, as previously described[39]. This procedure is described in full in Supplementary Method 4. The final output is a 2D light-sheet radiation-pressure force profile $F_{rad}(x, z)$.

Another modification from the previously described method was the implementation of a rapid automated algorithm based on the Radon transform to extract depth-resolved bead axial velocity and acceleration from the space-time bead trajectory images. The automated algorithm significantly reduced the time and labour that would otherwise be required to perform the coarse manual bead trajectory tracking over the entire 2D BM-mode datasets (200–400 beads total). This was essential to streamline the routine system alignment and force measurement procedure prior to each experiment. The full description of the Radon-transform-based automated force measurement algorithm is provided in Supplementary Method 5.

**LS-pfOCE data acquisition procedure**. Volumetric imaging with LS-pfOCE adopted a 3D BM-mode acquisition scheme[36], with the modification that slow-axis scanning was accomplished by translating the sample with a motorised stage (instead of galvanometer scanning as typically done for 3D-OCT data acquisition). For all LS-pfOCE results presented here, a BM-mode frame rate of 425 Hz was implemented with a radiation-pressure force modulation frequency of 20 Hz, which ensured sufficient temporal sampling per modulation cycle to reliably measure the phase shift w.r.t. the drive waveform of the bead response. The fast-axis scan range was kept at 90 μm to match the width (long axis) of the light sheet. A total of 6,400 BM-mode frames were acquired at each slow-axis position, supporting a theoretical (SNR-limited shot-noise)[36,67] and experimental displacement sensitivity of 27 pm and 76 pm at an OCT SNR of 28 dB, respectively. The lateral pixel size was 0.75 μm × 0.75 μm, which ensured that each probe bead (diameter of 1.7-μm for polystyrene and 1.9-μm for melamine-resin) were sampled at multiple lateral pixels. These acquisition parameters result in an acquisition time of 15 s per slow-axis position, which would total to 117 min for a slow-axis scan range of 350 μm implemented in Fig. 3a–c. However, due to the delayed stabilisation after stepping of our current motorised actuator (Thorlabs, ZST225B), we implemented a wait time of 6 s before initiating the acquisition at each slow-axis position. This significant wait time would not be necessary for a high-fidelity stepper motor. All instrument control and synchronisation were accomplished with a custom Lab-VIEW (2014 64-bit version) acquisition software.

**LS-pfOCE reconstruction procedure**. The reconstruction of micromechanical properties of the medium in the vicinity of each probe bead from the raw 3D BM-mode LS-pfOCE data followed a similar approach to quantifying micromechanical properties via Gaussian-beam PF-OCE[37], with modifications to implement the processing routine at multiple spatial voxels in the volumetric datasets (as opposed to 1D M-mode datasets in ref. [37].) and accommodate for the excitation by the light-sheet (instead of Gaussian) pump beam. The full description of the LS-pfOCE reconstruction procedure is provided in Supplementary Method 1 together with a flowchart that outlines the full procedure (Supplementary Fig. 2). Briefly, the LS-pfOCE reconstruction procedure can be divided into the following six steps.

(1) OCT image reconstruction, which implemented Fourier-domain OCT reconstruction and computational image formation procedure[6] for defocus correction along the fast axis to obtain the complex 3D BM-mode OCT image, $\tilde{S}(x, y, z, t)$.

(2) Phase-sensitive OCE reconstruction, which follows the procedure in ref. [37]. to extract the optical path length (OPL) response from the phase of $\tilde{S}(x, y, z, t)$, after phase registration. The OPL response as a function of time at each spatial voxel is described as a complex phasor with amplitude $A(x, y, z)$ and phase shift $\varphi(x, y, z)$ w.r.t. the drive waveform (for brevity, subsequent mention of "phase shift" shall refer to the phase shift w.r.t. the drive waveform).

(3) Image segmentation, which assigned the OPL response at each spatial voxel to either the PT response or the total response data region based on OCT image magnitude thresholds (values provided in Supplementary Method 2) and data exclusion criteria (indicated on Supplementary Fig. 2)[37].

(4) Photothermal (PT) response reconstruction, which reconstructed the 2D PT response amplitude $A_{PT}(x, z)$ and phase shift $\varphi_{PT}(x, z)$ from the OPL response at the spatial voxels assigned to the PT response data region in step 3.

(5) Isolation of the bead mechanical response, which subtracts the PT response reconstructed in step 4 from the OPL response at the spatial voxels assigned to the total response data region to isolate the mechanical response with amplitude $A_{mech}(x, y, z)$ and phase shift $\varphi_{mech}(x, y, z)$ at each spatial voxel. Then, the individual spatial voxels were clustered into their respective beads to compute the median mechanical response of each bead $A_{mech}(\mathbf{r}_c)$ and $\varphi_{mech}(\mathbf{r}_c)$, where $\mathbf{r}_c = (x_c, y_c, z_c)$ denotes the spatial coordinates of the bead centroid.

(6) Complex shear modulus reconstruction, which solved the 1D axial equation of motion of a sphere forced into an oscillatory motion, described by an amplitude $A_{mech}(\mathbf{r}_c)$ and phase shift $\varphi_{mech}(\mathbf{r}_c)$, in a viscoelastic medium by an external harmonically-varying force with amplitude $F_{rad}(\mathbf{r}_c)$, extracted from the measured force profile $F_{rad}(x, z)$. The complex shear modulus of the medium in the vicinity of each bead $G^*(\mathbf{r}_c)$ is given by:

$$\tilde{G}_{eff}(\mathbf{r}_c) = \frac{F_{rad}(\mathbf{r}_c) + m\omega^2[A_{mech}(\mathbf{r}_c)\exp(i\varphi_{mech}(\mathbf{r}_c))]}{6\pi a[A_{mech}(\mathbf{r}_c)\exp(i\varphi_{mech}(\mathbf{r}_c))]}, \quad (1)$$

where $\omega$, $m$, and $a$ denote the angular modulation frequency, mass, and radius of the oscillating sphere, respectively. The parameter $\tilde{G}_{eff}(\mathbf{r}_c)$ is a function of $G^*(\mathbf{r}_c)$ and is given by Oestreicher's model of an oscillating sphere in a viscoelastic medium, described by a complex shear wave number $k^*$, according to:

$$\tilde{G}_{eff}(\mathbf{r}_c) = G^*(\mathbf{r}_c)\left[1 - ik^*(\mathbf{r}_c)a - \tfrac{1}{9}k^{*2}(\mathbf{r}_c)a^2\right], \quad (2)$$

which can be inverted to obtain $G^*(\mathbf{r}_c)$.

All data processing was implemented in MATLAB 2017a.

Compared to the previous Gaussian-beam implementation of PF-OCE[36,37], the use of light-sheet radiation-pressure force excitation has the largest impact on steps 4 and 6 listed above. This is due to the non-uniform lateral intensity profile of the light sheet along its long axis (Fig. 1c, d), which causes both the radiation-pressure force and the PT response to vary not only as a function of depth (axial) but also across the fast axis (lateral along the long axis of light sheet). Supplementary Method 4 provides a full description of the LS-pfOCE force reconstruction procedure. For the PT response, another important distinction implemented here compared to our previous work[36,37] was that no exogenous PT reporters were added into the sample. This was done to improve the biocompatibility of LS-pfOCE for mechanobiological applications with live-cell imaging studies. The LS-pfOCE experiments presented here relied entirely on the intrinsic OCT scattering signal of the medium and the cumulative PT response measured at the coverslip (see Supplementary Fig. 3 the sample configuration) to reconstruct the full 2D profiles of the PT response amplitude and phase shift (Supplementary Fig. 4). Supplementary Method 3 provides a full description of the PT response calibration procedure for PAAm gels (Figs. 1e and 2a, b, d), collagen matrices (Fig. 2), and live-cell imaging in cell-seeded fibrin constructs (Fig. 3).

**Sample preparation**. All samples were prepared in #1.5 glass-bottomed petri dishes (MatTek, P35G-1.5-10-C), where the OCT beam interrogated the sample through the glass bottom (Supplementary Fig. 3). For the cell-free samples (i.e., PAAm gels and collagen matrices), a coverslip (Fisherbrand, 12540A) was placed on top of the sample well to provide measurements of the cumulative PT

responses[36,37]. For live-cell imaging, the sample petri dish was placed in a stage-top incubated bio-chamber (Okolab, UNO-PLUS) for environmental control.

PAAm gels (Figs. 1e and 2a, b, d) were prepared as previously described[37]. Briefly, acrylamide monomer (Bio-Rad, 40% Acrylamide Solution, 1610140), bis-acrylamide crosslinker (Bio-Rad, 2% Bis Solution, 1610142), and deionized water were mixed at appropriate concentrations (see Table 1 in ref. [37]. for the compositions of 3T1C and 3T2C gels). An aqueous suspension of 1.7-μm diameter polystyrene beads (Spherotech, PP-15-10) was added to the mixture at a concentration of 30 μL/mL to achieve a mean particle separation of 12 μm. The solution was mixed and desiccated for 15 min. PAAm polymerisation was activated with 10% ammonium persulfate (Bio-Rad, APS, 1610700) as the redox initiator and tetramethylethylenediamine (Bio-Rad, TEMED, 1610800) as the catalyst. 10% APS and TEMED were added at concentrations of 10 μL/mL and 1 μL/mL, respectively. Hydrogels were allowed to polymerise for 60 min prior to LS-pfOCE measurements.

Collagen matrices (Fig. 2c) were prepared with rat tail type I collagen (Corning, 354236) at a final collagen concentration of 2.0 mg/mL. An aqueous suspension of 1.9-μm diameter carboxyl-functionalized melamine-resin beads (microParticles GmbH, MF-COOH-S1000) was added to the mixture at a concentration of 19 μL/mL to achieve a mean particle separation of 14 μm. Melamine-resin beads were used here instead of polystyrene beads because they generated larger radiation-pressure force magnitude due to the higher refractive index. Carboxyl functionalisation ensured that the probe beads adhered to the collagen matrix rather than freely floating inside the pores. Three polymerisation protocols were implemented to form collagen matrices with different fibre architecture. The C1 sample was polymerised at 37 °C for 45 min to form a relatively uniform distribution of fine collagen fibre network. The C2 sample was polymerised at 22 °C (room temperature) for 45 min to promote formation of thicker collagen fibres with more heterogeneous distribution. The C3 sample was polymerised at 4 °C for 15 min, 20 °C for 15 min, then 37 °C for 15 min to form the most heterogeneous fibre architecture with the thickest collagen fibres[6]. For confocal reflectance microscopy in Fig. 2c, the three samples are prepared following the same protocol but without the addition of beads, since strong reflection from the beads would overpower the signals from the collagen fibres and saturate the detector.

Cell-seeded fibrin constructs consisted of NIH-3T3 fibroblasts (ATCC, CRL-1658) encapsulated in 3D fibrin hydrogels. The cells were maintained in tissue culture flasks with media consisting of Dulbecco's Modified Eagle Medium (DMEM, Life Technologies, 11965092) supplemented with 10% foetal bovine serum (FBS, Life Technologies, 16170086) and 1% penicillin-streptomycin (PS, Life Technologies, 15140122). Fibrin hydrogels were prepared with bovine fibrinogen (Sigma-Aldrich, F8630) reconstituted in DMEM without serum and antibiotics, at a final concentration of 2.5 mg/mL. NIH-3T3 cells and 1.9-μm diameter carboxyl-functionalized melamine-resin beads (microParticles GmbH, MF-COOH-S1000) were added to the sterile-filtered fibrinogen solution to achieve a final cell density of $7 \times 10^4$ cells/mL and an average bead spacing of 20 μm, respectively. Lower bead density was utilized here compared to in the collagen matrices to keep the volume fraction of exogenous beads below $1 \times 10^{-4}$ for live-cell imaging. The mixture was deposited into the sample well of a glass-bottomed petri dish, where polymerisation into fibrin was initiated by 2 U of bovine thrombin (Sigma-Aldrich, T4648) pre-aliquoted into the well. The cell-seeded constructs were incubated at room temperature for 5 min followed by 37 °C for 25 min, then, rehydrated with DMEM containing 10% FBS and 1% PS. The LS-pfOCE measurements were performed after 12 hr of incubation at 37 °C.

**Live-cell imaging protocols**. The cell-seeded fibrin constructs were incubated for 12 h prior to the LS-pfOCE measurements to allow for the cells to naturally spread. In the post-CytoD treatment condition, 50 μL of a 100-μM Cytochalasin D (Sigma-Aldrich, C2618) dissolved in DMSO was administered to the cell-seeded 3D fibrin construct and incubated at 37 °C for another 2 h. Immediately before LS-pfOCE measurements, the sample petri dish was placed inside the stage-mounted incubated bio-chamber. For 3D LS-pfOCE measurements (Fig. 3a–f), a cell with an elongated morphology and roughly oriented along the slow-axis scanning direction was selected for imaging. Guided by a real-time display of the *en face* OCT image, the cell was roughly positioned at the centre of the 80 μm × 350 μm (fast axis × slow axis) transverse FOV by translating the sample stage. This FOV allowed the spatial variation in the pericellular space to be captured around the cell, with an extended distance on either side of the cell tips (i.e., major elongated axis of the cell).

For dynamic monitoring of the response to CytoD treatment (Fig. 3g–i), time-lapsed LS-pfOCE was performed 15 min after adding 20 μL of the 100-μM CytoD to the dish. An elongated cell was likewise located, but instead of positioning the cell at the centre of the FOV, a tip of the cell was positioned adjacent to an 80 μm × 26 μm (fast axis × slow axis) transverse FOV along the slow axis. This FOV allowed the spatiotemporal variations in the pericellular space adjacent to the cell tip to be captured every 15 min while the CytoD treatment was taking effect. Two control experiments were performed. For the blank control with CytoD (Supplementary Fig. 9a), 3D fibrin constructs were prepared as described above but without seeding the cells. First, a 3D LS-pfOCE dataset was acquired, then, the CytoD treatment was added (50 μL of a 100-μM CytoD dissolved in DMSO) before another 3D LS-pfOCE dataset was acquired at approximately the same location. For the cell

control with DMSO (Supplementary Fig. 9b), a 3D LS-pfOCE dataset was acquired around a cell as described above. Then, 50 μL of pure DMSO (without CytoD) was added and the sample was left in the bio-chamber for 2 hr before another 3D LS-pfOCE dataset was acquired around the same cell.

**Parallel-plate shear rheometry of polyacrylamide gels**. Bulk complex shear modulus of each PAAm gel in Fig. 1e was measured with a parallel-plate shear rheometer (TA Instruments, DHR-3) using a 20-mm diameter plate geometry, as described in ref. [37]. Polymerisation was achieved directly on the rheometer plate by pipetting 200 μL of the activated polymer solution (i.e., after adding 10% APS and TEMED) onto the bottom plate. The gap was set 500 μm and the excess polymer solution was carefully removed. The sample was sealed on the side with mineral oil to prevent evaporation. Polymerisation was monitored in a time-sweep oscillatory test at oscillation frequency of 1 rad/s and shear strain of 0.5%. All samples were left to polymerise for 60 minutes, during which the stabilisation of shear moduli was confirmed. Then, a frequency-sweep oscillatory test was performed at the oscillation frequency ranging from 1–50 Hz and applied torque of 10 μN m. A total of 5 measurements were made for each PAAm concentration.

**Confocal reflectance microscopy of collagen matrices**. The confocal reflectance images in Fig. 2c were obtained with a Zeiss LSM 710 laser-scanning confocal microscope system, operating in the reflection mode at 488 nm with a C-Apochromat 40x/1.2 NA water-immersion objective. The images were taken at a transverse FOV of 106 μm × 106 μm with transverse resolution of 250 nm. Each image was averaged over 16 *en face* planes. The collagen fibrous structures were then segmented by thresholding at 80 percentile of pixel values in each image. To further enhance the collagen contrast, the processed images were gamma-corrected with a coefficient of 0.4. All shown confocal images share the same grey colour scale. Image data was processed in Python v3.7.7.

**Analysis of micromechanical heterogeneity in collagen matrices**. The micromechanical heterogeneity measured in sample C3 by LS-pfOCE was correlated to the microstructural architecture of the collagen matrix in Fig. 2e. Four metrics describing the local collagen matrix characteristics around each bead were obtained from the en face OCT image of sample C3. First, the probe beads were removed from the 3D-OCT image via a magnitude-based segmentation and dilation of the binary mask. Then, spatial voxels that were located ≤3 μm distance away from the nearest circumference of each bead (i.e., a local volume of matrix within the microenvironment of each bead) were collected. Three bead-wise statistical metrics were computed from the collection of spatial voxels around each bead, providing the first 3 metrics in the list below. Lastly, we performed resolution enhancement on the OCT image via coherent-averaging across BM-mode frames and a 1.6× computation bandwidth expansion, as previously described[68]. The resolution enhancement improved the ability to discern individual collagen fibres, allowing for the measurement of the last metric.

(1) Fibre thickness: due to the limited OCT image resolution, a direct measurement of collagen fibre thickness was not feasible. Instead, we computed the 0.95 quantile (Q95) of OCT intensity as a surrogate measure for maximum collagen fibre thickness at each bead, under the premise that a thicker fibre generates higher OCT scattering intensity.
(2) Fibre content: we computed mean OCT intensity (Mean) as a surrogate measure of the overall collagen content around each bead, under the premise that only the collagen matrix would generate OCT scattering signal.
(3) Fibre volume fraction: we first segmented the OCT image based on a threshold 6 dB ≥ OCT SNR ≥ 14 dB, where 6 dB was considered a minimum SNR to be above noise and 14 dB was observed as the minimum SNR for a collagen fibre. Then, the fraction of spatial voxels that passed the threshold were computed as the local collagen fibre volume fraction around each bead.
(4) Fibre connectivity: each bead was located on the resolution-enhanced image and the number of fibre branches that were connected to each bead (representing a node in the fibre network) was manually counted. The counter was blinded to the LS-pfOCE results during the counting process.

Spearman's rank correlation coefficients and p-values between the bead-wise $G'$ and $R$ measured by LS-pfOCE and the three local image metrics are indicated on Fig. 2e. (Although linear fit lines were plotted to guide the visualisation of the trends, the reported correlation results represent the Spearmans rank correlation and not the linear regression.) All analysis was implemented in MATLAB 2017a.

**Analysis of cell-mediated spatiotemporal variations in ECM micromechanical properties**. For the analysis of spatial variations in the micromechanical properties in the pericellular space (Fig. 3f), the distance $r$ from each probe bead to the cell body was defined as the length of the shortest line connecting the centroid of the bead to a point on the surface of the cell. The cell body (i.e., white cell structure in Fig. 3a–c) was segmented from the 3D BM-mode image with a previously described method based on temporal speckle contrast to distinguish the dynamic cellular structures from the static fibrin background[6,58,59] (Here, standard deviation of OCT magnitude image was computed across temporal BM-mode frames instead of the "burst" of 8 OCT

volumes in refs. [6,58,59].). To calculate $G'$ and $R$ as a function of $r$, the beads were grouped based on the distance $r$, with each group spanning 3 μm. The error bars were calculated as the 95% confidence interval for the mean value in each distance group with Student's $t$-distribution. The curve fit in the inset of Fig. 3f was conducted by a least squares polynomial fit over the domain of $r \leq 30$ μm. This portion of the analysis was implemented in Python v3.7.7.

For time-lapse monitoring immediately after adding the CytoD treatment, the cumulative bead displacement $S$ reported on Fig. 3g–i was computed from the spatial coordinates of the bead centroid $\mathbf{r}_c = (x_c, y_c, z_c)$ at each time point, after correcting for the bulk sample shift. The $x$ component of the cumulative displacement of the $m^{th}$ bead at the $n^{th}$ time point was computed from:

$$\Delta x_c(t_n; m) = \sum_{i=2}^{n}[[x_c(t_i; m) - x_c(t_{i-1}; m)] - \frac{1}{N}\sum_{j}^{N}[x_c(t_i; j) - x_c(t_{i-1}; j)]], \quad (3)$$

where $N$ denotes the total number of beads in the field of view. The inner summation term represents the bulk shift, estimated as the mean displacement of all beads between each adjacent time points. The $y$ and $z$ components of the cumulative bead displacement were computed in the same manner. The signed cumulative displacement of the $m^{th}$ bead at the $n^{th}$ time point was computed from the displacement vector $\Delta\mathbf{r}_c = (\Delta x_c, \Delta y_c, \Delta z_c)$ by:

$$S(t_n; m) = \text{sgn}[\Delta_{\text{major}}(t_n; m)]\|\Delta\mathbf{r}_c(t_n; m)\|, \quad (4)$$

where $\text{sgn}[\Delta_{\text{major}}]$ denotes the sign of the major $\Delta\mathbf{r}_c$ component (i.e., $\Delta x_c$, $\Delta y_c$ or $\Delta z_c$), defined as the largest component of the directional displacement vector, $(|\Delta x_c|, |\Delta y_c|, |\Delta z_c|)/\|\Delta\mathbf{r}_c\|$, averaged across all beads and time points. The beads were divided into three groups based on their $S$ values at the last time point to visualise the temporal variations of $G'$ and $R$ in Fig. 3g–i. This portion of the analysis was implemented in MATLAB 2017a.

**Statistical analysis**. For PAAm gels, two-sided Welch's $t$-test was implemented to test the difference between group means of LS-pfOCE measurements and shear rheometry measurements in both 3T1C and 3T2C PAAm gels in Fig. 1e. For collagen matrices, two-tail Spearman rank correlation was computed to test the correlation between LS-pfOCE measurements and local OCT image metrics in sample C3. The correlation was considered significant at the 95% confidence level; the $p$-values are reported on Fig. 2e. For the 3D live-cell imaging study, two-sided Welch's $t$-test for equality of group means and Levene's test for equality of variances were implemented to test the difference in group means and variances between 3D LS-pfOCE measurements under the normal and CytoD conditions, respectively. The difference was considered significant at the 95% confidence level; the $p$ values are reported in Fig. 3e. For time-lapsed dynamic monitoring of response to CytoD, two-tail Pearson linear correlation was computed between cumulative change in LS-pfOCE measurements and cumulative bead displacements. The correlation was considered significant at the 95% confidence level; the $p$ values are reported on Fig. 3g. In all boxplots, including Supplementary figures, horizontal line indicates median, box spans 1st to 3rd quartile, and whisker length corresponds to 1.5 times the interquartile range. All statistical tests were implemented in Python v3.7.7 except the analysis of correlations, which were implemented in MATLAB 2017a.

**Reporting summary**. Further information on research design is available in the Nature Research Reporting Summary linked to this article.

## Data availability
Data used to generate the plots in this manuscript are available at Harvard Dataverse (https://doi.org/10.7910/DVN/IJXI4P). Raw and other primary processed data generated in this study are available from the corresponding authors upon reasonable request.

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

## Acknowledgements

This work made use of the Cornell Center for Materials Research Shared Facilities which are supported through the NSF MRSEC programme (DMR-1719875). This work was conducted in part at the Cornell University Biotechnology Resource Center (NIH 1S10RR025502-01) for data collected on the Zeiss LSM 710 laser-scanning confocal fluorescence microscope. Y.L., N.L. and S.G.A. have been supported in part from both NIBIB-R21EB024747 and NIGMS-R01GM132823. J.C.L. was supported from NIGMS-R01GM132823.

## Author contributions

Y.L. and S.G.A conceived the light-sheet idea and initiated the project. Y.L. and N.L. constructed the LS-pfOCE system. Y.L. conducted all LS-pfOCE measurements. N.L. developed LS-pfOCE reconstruction algorithms and processed all data. Y.L., N.L. and J.C.L. contributed to analysis of primary processed data and presentation of results. J.C.L. maintained cell cultures, prepared and imaged collagen matrices, and fabricated fibrin constructs for the live-cell imaging study. Y.L. and N.L. wrote the manuscript. S.G.A supervised the project. All authors helped revise the manuscript.

## Competing interests

N.L. and S.G.A. are listed as inventors on U.S. Patent No. US10072920B2 and US10197379B2. Y.L., N.L., and S.G.A are listed as inventors on U.S. Provisional Patent Application No. 62/968,961. The remaining authors declare no competing interests.
