## [Peer Review File · Nature Communications]

Light-sheet photonic force optical coherence elastography for high-throughput quantitative 3D micromechanical imagingREVIEWER COMMENTS

Reviewer #1 (Remarks to the Author):

This manuscript reports a new method for active microrheological probing of soft materials using a light sheet to simultaneously optically manipulate all microbeads in the field of view. The authors validate this approach using (linear elastic) polyacrylamide gels, fibrous collagen gels, and to visualize fibroblast-mediated deformations. The validation of this technique is carefully implemented and shows good agreement with bulk microrheology. However, I note that these case studies are relatively established experimental systems and there is limited fundamental or mechanistic insight into soft materials or mechanobiology. I believe there is significant promise to this technology, which is highly innovative and potentially quite powerful. However, a general audience may not fully appreciate the novelty of this approach, without additional proofs of concept that are non-trivial to implement using existing techniques.

In particular, this could be a very powerful way to directly measure local spatial heterogeneity in soft materials, which cannot be resolved using bulk rheology. I think this technique could very easily measure locally nonuniform stiffness in a gradient crosslinked polyacrylamide gel (e.g. Engler's work) or across the interface of two collagen gels of differing concentration (e.g. Reinhart-King). I also believe there is quite interesting structural information about network heterogeneity encoded in Fig 2d that is getting short shrift in the manuscript (see also PMID: 28719577), considering both microstructure (from OCT) and local microrheology can be directly measured. Combining these measurements with various existing fiber network models could be of great interest to the soft materials community.

Similarly, inhibiting cell contractility with cytochalasin D and measuring the resulting matrix relaxation (Fig 3) is a standard control for traction measurements. It would be very exciting to track the irreversible, plastic deformations of the collagen matrix at longer times driven by fibroblasts, again combining OCT measurements of microstructure with local measurements of stiffness. This would build nicely on the results of ref 33, which reports stress stiffening and includes a theoretical model, but required laborious bead by bead microrheology measurements with an optical tweezer.

Reviewer #2 (Remarks to the Author):

The paper entitled "Light-sheet photonic force optical coherence elastography for high-throughput quantitative 3D Micromechanical imaging" by Yuechuan Lin et al combines quantitative 3D imaging of extracellular matrix mechanics with cellular scale resolution with simultaneous dynamic monitoring of cell-mediated changes to pericellular viscoelasticity. This is demonstrated in the current paper by using light-sheet photonic force coherence elastography (LS-pfOCE). The demonstration is done by imaging the micromechanical heterogeneity of 3D collagen matrices and life cell study micromechanical heterogeneity induced by NIH-3T3 cells seeded in 3D fibrin constructs. The authors also demonstrate using LS-pfOCE enables they are able to quantify temporal variations in pericellular viscoelasticity when the cell is exposed to a drug that alters cellular activity. In this way it is demonstrated that 4D spatiotemporal variations in micromechanics of the cell can be studied in detail.

I believe that this is rather significant contribution to the field. The cited literature is relevant and well chosen. I have some difficulties with understanding some details in the section "Characterisation and validation of the LS-pfOCE system", especially fig 1 e. Firstly when measuring G' and G'' in PAAm gels measured with 3D LS-pfOCE and bulk shear rheometer – why were the specific values of the number of the beads chosen? Secondly – this result does not look

very convincing. Any comments? Fig 1c – would it be possible to conduct measurements for larger span of lateral axis? There seems to be some consistent deviation from the GLMT result as compared to the measurement for these values of lateral axis so the equation is whether these results show just the size of the error or is that a trend.

Fig 3 – box plots of G' and R – what am I supposed to be convinced about by looking at these results? Nothing much happens in these curves with time for the three conditions. Is that what I am supposed to conclude, or is there some more profound result that I cannot see. I am supposed to be convinced that this imaging of the live cells provides new otherwise inaccessible data? In the discussion the authors are talking about more sophisticated light-sheet generation optics to introduce laterally uniform light sheet PF excitation with extended depth of focus. I would have thought that you need laterally uniform light sheet to be able to draw conclusions that are made in this paper.

The methodology is sound but I would like to see a bit more discussion (and comparison with other methods) about the ultimate resolution of these methods and discussion what is needed in practice and are we there yet or not.

I think that it will be interesting to have this paper presented to the wider readership but maybe after addressing some points made above.

Reviewer #3 (Remarks to the Author):

The authors present a method for using a sheet of light to impose forces on beads embedded within a hydrogel or ECM mimic. Using this system and an OCT technique to measure bead displacement, the authors are able to extract the complex shear modulus from many beads in short acquisition times. This technique addresses many shortcomings with current approaches, but key data from the manuscript are difficult to interpret as presented, thus dampening enthusiasm for the manuscript.

- Overall, the figures are difficult to comprehend with subpanel labels in different positions for different subpanels and many different fonts, including overlapping text (Fig. 2).

- In Fig. 2c, an indication, perhaps with open circles, on the confocal reflectance image of where the bead measurements were taken would be very helpful for contextualizing the results shown below.

- Further, in Figure 2, the data for fiber microarchitectural properties are very similar, particularly for fiber thickness and heterogeneity. These plots seem to nearly overlay 1:1. How was heterogeneity calculated? Is there a reason that heterogeneity and fiber diameter should be so similar?

- The effects of CytoD are difficult to interpret. In Fig 3b-c, it appears that treatment with CytoD leads to a reduction in G' , and the FWHM data written on Fig. 3e supports this observation. However, the violin plot shows a clear increase in G' with CytoD treatment. What explains this discrepancy?

o Additionally, the CytoD treatment is compared to a 'Normal' condition – is 'Normal' a DMSO load control? The reference to the supplemental data suggests that there is not DMSO in these samples, but that should be clarified in the caption.

- The observed decay in elasticity as a function of distance from the cell surface (Fig. 3f, Fig. S10, etc) is attributed in the text to cellular remodeling of the pericellular ECM. However, it is expected that the presence of the cell would impact these measurements. For example, one would expect a similar decay in elasticity if a stiff sphere was embedded within the matrix, regardless of matrix remodeling. Beads near the sphere-ECM interface would displace by a lesser degree as the sphere stiffness increased, even if the surrounding ECM remained at constant G' and G'' . How do the changing mechanics of the cell, in response to CytoD or more broadly as cells spread and change shape, affect measurements in the matrix independent of changes to the matrix due to matrix deposition or proteolysis?

We graciously thank all referees for reviewing our manuscript. Your constructive comments and suggestions have significantly helped us in revising the manuscript. Our point-by-point responses to the referees are discussed below, with changes to manuscript highlighted in red text.

Reviewer #1 (Remarks to the Author):

This manuscript reports a new method for active microrheological probing of soft materials using a light sheet to simultaneously optically manipulate all microbeads in the field of view. The authors validate this approach using (linear elastic) polyacrylamide gels, fibrous collagen gels, and to visualize fibroblast-mediated deformations. The validation of this technique is carefully implemented and shows good agreement with bulk macrorheology. However, I note that these case studies are relatively established experimental systems and there is limited fundamental or mechanistic insight into soft materials or mechanobiology. I believe there is significant promise to this technology, which is highly innovative and potentially quite powerful. However, a general audience may not fully appreciate the novelty of this approach, without additional proofs of concept that are non-trivial to implement using existing techniques.

We appreciate the referee's thoughtful comments and for acknowledging that our technology "is highly innovative and potentially quite powerful". For a proof-of-concept demonstration of our technology, as the reviewer pointed out, the reported experimental systems in this manuscript are relatively established in order to demonstrate the unprecedented mechanical imaging capabilities of LS-pfOCE for readers that work in similar application areas (*e.g.*, polymer micromechanics and cell mechanobiology, as the reviewer also discussed below) can easily relate to. Our technology demonstrated unique volumetric mechanical imaging capabilities inside scattering viscoelastic media over a large depth range without requiring any additional time-consuming optical alignment during measurements. These mechanical imaging capabilities have never been achieved by any other existing competing technologies, which underpins the significance and novelty of our method. (For instance, it is not trivial—but extremely laborious, as the referee later pointed out—to reproduce our 3D dynamic live-cell imaging results in Fig. 3 with conventional optical tweezer-based techniques.)

In addition to demonstrating unique imaging capabilities in "familiar" settings, we were also able to demonstrate new analysis and glean information that has never been reported by existing techniques in the same experimental systems (*e.g.*, the correlation between collagen micromechanics and microarchitecture in Fig. 2e, and measurements of time-dependent stiffening and softening of pericellular viscoelasticity following cytochalasin D treatment that are correlated to cell-induced matrix deformation in Fig. 3g–i). In this manuscript, we want to establish and emphasize new capabilities of our technology and its accessibility. Indeed, this manuscript forms the basis for more profound fundamental mechanobiological questions to be further investigated in the future that will fully exploit the advantages of our technology, as suggested by referee. **We include this discussion in line 327-333 of manuscript.**

In particular, this could be a very powerful way to directly measure local spatial heterogeneity in soft materials, which cannot be resolved using bulk rheology. I think this technique could very easily measure locally nonuniform stiffness in a gradient crosslinked polyacrylamide gel (*e.g.* Engler's work) or across the interface of two collagen gels of differing concentration (*e.g.* Reinhart-King).

We thank the referee for pointing out one of the crucial advantages of LS-pfOCE that "could be a very powerful way to directly measure local spatial heterogeneity in soft materials". We strongly

agree with the referee's comment that our technology should be very promising at quantifying local "nonuniform stiffness" in both gradient cross-linked PAAm and across the interface of two soft materials with different mechanics. In fact, our previous publication (Ref. 36) has already demonstrated the ability to resolve mechanical contrast across the interfaces of two hydrogels with different mechanical properties (Fig. 6 in Ref. 36). In this previous work, we only showed the mechanical response (*e.g.*, vibration amplitude and phase delay) of the probing bead. Since the reconstruction of the complex shear modulus from the bead-wise mechanical response is linear on each side of the homogeneous samples, we should straightforwardly expect that LS-pfOCE is able to quantify such a sample as both techniques share the same fundamental operating principle. In addition, the results in our manuscript shown in Fig. 2a further illustrates that LS-pfOCE is capable of measuring the local shear modulus in both homogeneous and heterogeneous hydrogels.

I also believe there is quite interesting structural information about network heterogeneity encoded in Fig 2d that is getting short shrift in the manuscript (see also PMID: 28719577), considering both microstructure (from OCT) and local microrheology can be directly measured. Combining these measurements with various existing fibre network models could be of great interest to the soft materials community.

We agree with the referee's insightful suggestion that our measured local micromechanical properties of hydrogels could be correlated to the local fibre network characteristics. Indeed, we attempted to glean insights into the correlation between network micromechanics and microarchitecture from the scatter plots in Fig. 2e. Motivated by the referee's suggestion to delve deeper into existing fibre network models, we have now expanded upon our original analysis of 3 local OCT scattering statistics (Q95, Mean, and Std) as surrogates for collagen architecture. To accomplish this we first leveraged our recently published technique for resolution enhancement of the OCT images (Ref. 42), to improve the ability to discern collagen fibres in Fig. 2c. With resolution-enhanced OCT images, we computed 2 additional metrics: local collagen volume fraction in the vicinity of each bead and number of connected fibre branches at each bead, both of which have significant implications for a lattice-based model of fibre network and network rigidity percolation. We found significant correlation between both fibre network metrics and our micromechanical measurements. **We have added new sub-figures in Fig. 2e in the manuscript and discussed these findings in lines 203-208.** A more rigorous theoretical model of such correlation requires a strong background in materials science and mathematical modelling, which is out of the scope of this manuscript. On the other hand, our technology has the potential to help inform and validate the development of computational models in materials science by providing experimental quantification of the micromechanical properties of biopolymer constructs. **We added this discussion in our manuscript from lines 390-393.**

Similarly, inhibiting cell contractility with cytochalasin D and measuring the resulting matrix relaxation (Fig 3) is a standard control for traction measurements. It would be very exciting to track the irreversible, plastic deformations of the collagen matrix at longer times driven by fibroblasts, again combining OCT measurements of microstructure with local measurements of stiffness. This would build nicely on the results of ref 33, which reports stress stiffening and includes a theoretical model, but required laborious bead by bead microrheology measurements with an optical tweezer.

We agree with the reviewer's comment that treatment with cytochalasin D is considered as a standard control to induce cell relaxation in traction experiments. The reason we followed this standard approach is that by doing so, our measurements can be compared with the published results of other well establish techniques. This allows us to draw connections between our

unprecedented 3D dynamic ECM micromechanical mapping results and existing findings from optical tweezer-based and traction force studies. By conducting this standard experiment, we showed that LS-pfOCE not only provides information comparable to OT-based approaches *without* requiring laborious bead measurements, but also uncovers new interesting phenomena that have never been observed before, as shown in Fig. 3g-i in our manuscript. Here, we found that the micromechanical properties of the pericellular matrix of cytochalasin D treated cells does not simply follow a reduced trend in stiffness as cell contractility is inhibited, but surprisingly undergoes both softening *and* stiffening in a deformation-dependent manner! The latter has not been observed by OTs, but could be important in some mechanobiological studies (Refs. 49, 50).

We greatly appreciate the referee's comment of pointing out an exciting future/potential capability of LS-pfOCE to track plastically deformed matrix in pericellular regions. In fact, the traction force optical coherence microscopy (TF-OCM) developed by our group (Ref. 6) has already shown that there are residual and irreversible deformations induced by cell contraction (Fig. 6 in Ref. 6). Our findings, shown in Fig. 3f and Supplementary Fig. 10, also showed irreversible changes in the micromechanical properties of some pericellular regions over long durations due to cell-ECM interactions. In the future, we plan to combine TF-OCM and LS-pfOCE to perform additional novel and unique mechanobiological studies that not only track irreversible ECM deformations, but also investigate the correlation between cellular traction forces and associated cell morphology/invasion patterns. Furthermore, for traction force studies, LS-pfOCE has a great potential to facilitate a more accurate reconstruction of 3D cellular traction field by providing precise 4D substrate micromechanics as the input. **This discussion has been added in our manuscript in lines 381-384.**

Reviewer #2 (Remarks to the Author):

The paper entitled "Light-sheet photonic force optical coherence elastography for high-throughput quantitative 3D Micromechanical imaging" by Yuechuan Lin et al. combines quantitative 3D imaging of extracellular matrix mechanics with cellular scale resolution with simultaneous dynamic monitoring of cell-mediated changes to pericellular viscoelasticity. This is demonstrated in the current paper by using light-sheet photonic force coherence elastography (LS-pfOCE). The demonstration is done by imaging the micromechanical heterogeneity of 3D collagen matrices and life cell study micromechanical heterogeneity induced by NIH-3T3 cells seeded in 3D fibrin constructs. The authors also demonstrate using LS-pfOCE enables they are able to quantify temporal variations in pericellular viscoelasticity when the cell is exposed to a drug that alters cellular activity. In this way it is demonstrated that 4D spatiotemporal variations in micromechanics of the cell can be studied in detail. I believe that this is rather significant contribution to the field. The cited literature is relevant and well chosen.

We thank the referee for their review comments and appreciation of our technology as being "rather significant contribution to the field".

I have some difficulties with understanding some details in the section "Characterisation and validation of the LS-pfOCE system", especially fig 1 e. Firstly when measuring G' and G'' in PAAm gels measured with 3D LS-pfOCE and bulk shear rheometer – why were the specific values of the number of the beads chosen? Secondly – this result does not look very convincing. Any comments?

We thank the referee for indicating this potential point of confusion. The number of beads are not specifically chosen, but reported based on the amount of particles present in the volume acquired within the fibrous hydrogel. This quantity can be estimated when the average edge-to-edge spacing between particles, bead size, and dimensions of the volume are known. The benefit of having an increased number of beads enables higher spatial sampling of the local heterogeneous mechanics within fibrous hydrogels. However, an excessive number of particles may alter the micromechanical environment of the hydrogel construct or cellular behaviour. To optimize this trade-off, we chose an average particle spacing of 12-14 μm , which was also utilized in previous OT-AMR studies (Ref. 32, 33), and is consistent with bead densities reported in TFM studies (Ref. 6, 8).

The comparison between storage modulus G' as measured by LS-pfOCE and bulk shear rheometry statistically agrees with each other. However, as the referee pointed out, there is a significant difference in the loss modulus G'' . This discrepancy is expected and can be attributed to the distinct polymer mechanics at the micro- versus macro-scale. As discussed in our previous work (Ref. 37), our technology measures local micro-scale mechanics while bulk shear rheometry measures ensemble-averaged mechanics. In homogeneous hydrogels (*i.e.*, polymer network mesh size < bead size, such as PAAm measured in Fig. 1e), the local micro-scale stiffness (measured by our technology) should be consistent with ensemble-averaged macro-scale stiffness (measured by bulk shear rheometer) and, therefore, G' agrees between the two approaches. However, due to distinct viscous responses (*e.g.*, viscous drag of fluid flow through the porous polymer network) at the micro- versus macro-scale level, the measured G'' is significantly different between the two approaches, which corroborates with the results reported by OT-AMR (in Ref. 40, bulk measurements are in general insensitive to local micro-scale heterogeneities, whereas OT-AMR reveals significant variation across spatial locations). **We have also added this citation into our manuscript to support our argument in manuscript lines 130-131.**

Fig 1c – would it be possible to conduct measurements for larger span of lateral axis? There seems to be some consistent deviation from the GLMT result as compared to the measurement for these values of lateral axis so the equation is whether these results show just the size of the error or is that a trend.

The lateral extent of our force measurement in Fig. 1c was selected based on a combination of experimental design parameters: (1) the lateral span relevant to our volumetric LS-pfOCE measurement FOV, (2) the required frame rate to capture maximum bead acceleration at peak radiation pressure force, and (3) the necessary lateral sampling (*i.e.*, $\mu\text{m}/\text{pixel}$ A-scan spacing) to obtain the lateral force profile. Factor (1) depends on the long axis width of the light-sheet PF beam. For a given optical power of the PF beam, extending the lateral axis of the light-sheet (to increase the lateral span for LS-pfOCE measurements) will reduce the maximum force that can be applied to each bead, and, therefore, decrease the maximum measurable stiffness in the fibrous constructs. (Due to the intrinsic advantage of the light-sheet in having lower phototoxicity, a relatively larger span in the lateral axis can be achieved if a higher powered PF laser were implemented.) Factors (2) and (3) are related to scanning speed supported by our galvanometer. We decided to prioritize the quality of our force measurement (by ensuring sufficient spatial and temporal sampling) across the lateral span chosen for LS-pfOCE measurements over expanding/maximizing the lateral dimension of the light-sheet beyond what would be needed to reconstruct pericellular viscoelastic properties over the volume of interest.

The consistent deviation from the GLMT results might be attributed to aberrations from an imperfect optical system. Aberration can change the beam profile and also redistribute the optical power on the object plane, resulting in an asymmetric force profile and deviation from theoretical expectations, as discussed in our previous work where we introduced this OCT-based radiation pressure force measurement method (Ref. 39). Since we were unable to consider aberrations in our simulation, such a discrepancy should be expected. We have added explanations in our manuscript lines 100-101 and Supplementary Method 6 to address this question.

Fig 3 – box plots of G' and R – what am I supposed to be convinced about by looking at these results? Nothing much happens in these curves with time for the three conditions. Is that what I am supposed to conclude, or is there some more profound result that I cannot see. I am supposed to be convinced that this imaging of the live cells provides new otherwise inaccessible data?

The time-dependent trends in Fig. 3g–i demonstrate phenomena that has never been reported by other existing techniques that have been used to probe pericellular viscoelasticity. Current findings in the field demonstrate the pericellular matrix softening following inhibition of cellular contractility (Refs. 32-33). While we indeed observed this behaviour (consistent with previous results), our results also uncover a surprising and previously unobserved phenomenon—that pericellular matrix undergoes both softening *and* stiffening in a cell-mediated matrix deformation-dependent manner. As discussed in lines 286-301 of the manuscript, “regions of the matrix that experience a negative deformation tend to exhibit an initially lower G' (~100 Pa, similar to the native fibrin in the cell-free regions in Fig. 3d) that increases over time (Fig. 3g). Conversely, regions of the matrix that experience a positive deformation tend to exhibit an initially higher G' (>200 Pa, similar to the cell-modified pericellular fibrin in Fig. 3a, b) that decreases over time; this is also accompanied by an opposite trend in R (Fig. 3i)”. The softening *and* stiffening in pericellular regions have not been observed by OT-AMR and other existing approaches, but could be important in some mechanobiological studies (Refs. 49-50).

In the discussion the authors are talking about more sophisticated light-sheet generation optics to introduce laterally uniform light sheet PF excitation with extended depth of focus. I would have thought that you need laterally uniform light sheet to be able to draw conclusions that are made in this paper.

A uniform light-sheet force can benefit our technology with consistent maximum measurable stiffness over the whole field-of-view. However, a uniform light-sheet force is not necessary in our technology, as we have demonstrated. In order to make quantitative measurements of micromechanical properties, we simply require a calibration of the radiation pressure force profiles along the lateral and axial spans of the light-sheet PF beam. A laterally uniform light-sheet would provide optical forces distributed uniformly along the axis of the light-sheet, whereas in our study we specifically accounted for the non-uniform profile of optical force. Our calibration procedure performed a 2D force measurement. We accounted for the spatial variation in force profile when reconstructing shear modulus from the measured bead-wise mechanical response. We have clarified how the non-uniform light-sheet was accounted for in manuscript lines 338-341.

The methodology is sound but I would like to see a bit more discussion (and comparison with other methods) about the ultimate resolution of these methods and discussion what is needed in practice and are we there yet or not.

We thank the referee for suggesting the inclusion of this important discussion in our manuscript. We consider LS-pfOCE to be especially powerful for high-throughput volumetric mechanical imaging in scattering samples. With this scope, alternative methods include other optical coherence elastography techniques (Refs. 24-26). However, the resolution of existing OCE techniques are all on the order of hundreds of μm (for example in compression OCE, the axial spatial resolution is limited by the axial window length over which strain is calculated). From an imaging perspective, PF-OCE resolution is governed by the statistical sampling of randomly distributed probe beads, which in our experiments are 12-14 μm spacing. From the perspective of quantifying local micromechanical properties, G' and G'' measured at each bead location represents the average viscoelastic properties within the “microenvironment” surrounding each bead—extending as far as the *mechanical interaction length* of bead oscillation (Ref. 59). This is expected to be much smaller than our bead spacing (for a bead oscillation amplitude < 1 nm, the bead oscillation-induced deformation of the surrounding medium would rapidly drop below the displacement noise floor away from the bead, thus isolating our measurement essentially to the vicinity of each bead). Smaller beads can allow for smaller bead spacing (i.e., higher number of beads per volume can be used given the same limit on bead volume fraction) and better mechanical imaging resolution, however, at the cost of lower photonic forces (indicating lower maximum measurable stiffness of samples). In addition, our resolution is also limited by the OCT imaging resolution (the transverse resolution and axial resolution in air were 2.3 μm and 3.4 μm , respectively). In general, the achievable micromechanical imaging resolution of our LS-pfOCE is comparable with that of OT-AMR under similar conditions. To tailor the mechanical resolution to a specific application would require overall considerations of bead size, bead spacing, maximum available photonic force, optical resolution of imaging system and stiffness of sample-under-measurement. In practice, our current setup is designed to support micromechanical imaging of fibrous ECM constructs and live-cell imaging requiring cellular resolution. **We have added a discussion on the resolution of LS-pfOCE with comparisons to other existing techniques in manuscript lines 358-372.**

I think that it will be interesting to have this paper presented to the wider readership but maybe after addressing some points made above.

We greatly appreciate the referee’s comments and suggestions to improve our manuscript for a wider readership.

Reviewer #3 (Remarks to the Author):

The authors present a method for using a sheet of light to impose forces on beads embedded within a hydrogel or ECM mimic. Using this system and an OCT technique to measure bead displacement, the authors are able to extract the complex shear modulus from many beads in short acquisition times. This technique addresses many shortcomings with current approaches,

We thank the referee for their comments/suggestion and acknowledgment of LS-pfOCE in “addressing many shortcomings with current approaches”.

but key data from the manuscript are difficult to interpret as presented, thus dampening enthusiasm for the manuscript. - Overall, the figures are difficult to comprehend with subpanel labels in different positions for different subpanels and many different fonts, including overlapping text (Fig. 2).

We appreciate the referee for indicating this issue. In response, we have reformatted all figures in the main manuscript and supplement to improve their clarity and ease of interpretation. The changes include: increased spacing between subpanels and labels to avoid overlapping text, using the same font across all figures, and larger font size for plot legends where applicable. **To make our results Fig. 3f more straightforward for readers to interpret, we also normalized the measured micromechanics to their corresponding background per cell. The absolute value of our results is instead placed in Supplementary Data 2.**

- In Fig. 2c, an indication, perhaps with open circles, on the confocal reflectance image of where the bead measurements were taken would be very helpful for contextualizing the results shown below.

In Fig.2c, the confocal reflectance images (top row) are not co-registered with the OCE images (middle and bottom rows). Since confocal microscopy achieves finer optical resolution, the images shown serve only to demonstrate the varying collagen fibre architectures inside hydrogels we probed utilising LS-pfOCE. All collagen samples were fabricated following the same protocol for consistency—the only difference being whereas beads were added to the constructs for OCE imaging, the confocal microscopy samples have no beads added. This is because beads encapsulated in the collagen samples are highly scattering, leading to strong signal that renders collagen fibres difficult to detect and saturates the confocal microscope detector. **We have added this information to the caption of Fig. 2 and Methods section in lines 172-173 and 548-550, respectively.**

- Further, in Figure 2, the data for fibre microarchitectural properties are very similar, particularly for fibre thickness and heterogeneity. These plots seem to nearly overlay 1:1. How was heterogeneity calculated? Is there are reason that heterogeneity and fibre diameter should be so similar?

The referee points out an interesting detail. Firstly, we would like to clarify that the y-axis data in Fig. 2e were computed from the local OCT image statistics (Q95, Mean, and Std) within a 3- μ m radius of each bead (see Methods, Analysis of micromechanical heterogeneity in collagen matrices), and these image metrics represent *surrogates* for collagen fibre characteristics. This was because our OCT imaging resolution was not sufficient for analysis of the fibre structure (*e.g.*, with CT-FIRE software [Ref. 41]). Based on the referees' comments, we have improved the rigor of our analysis here by adding 2 new metrics—local collagen fibre volume fraction and connectivity at each bead—which have a more direct physical interpretation in relation to existing fibre network models in place of Std (our surrogates for heterogeneity). (See our response to Reviewer 1's comment on page 2) The implementation of new metrics was enabled by leveraging our recently published technique for resolution enhancement of the OCT image, which improve the ability to discern the collagen fibres in Fig. 2c. **We added a discussion of these results in relation to existing fibre network models in lines 187-211. We also clarified how each of the metrics were computed in the Methods section lines 614-632.** We believe the network heterogeneity is, in a way, the culmination of both fibre thickness and network connectivity. The presence of a thicker fibre would imply that more collagen is bundled together, leaving heterogeneous porous architecture in its vicinity (as opposed to a more uniform architecture in C1). Meanwhile, a higher degree of local network connectivity means that there are more branches or connected fibre segments at the analysed location, providing a more heterogeneous microenvironment.

To discern microarchitectures between the three collagen samples, we conducted fibre network analysis on the high-resolution confocal reflectance images using CT-FIRE software (Ref. 41). By comparing the distributions of fibre length and thickness, we found significantly different fibre microstructural properties between the three collagen samples. Meanwhile, the measured FWHMs of the distributions in collagen C2 and C3 are significantly larger than that of collagen C1, indicating more heterogeneity in C2 and C3. We also conduct two-dimensional autocorrelation of the three confocal images. The results show significant difference in average pore-sizes between the three collagen samples, which further show the different microarchitectural properties in three samples. **These new findings are added to the Supplementary Data 5 and discussed in manuscript lines 156-158.**

- The effects of CytoD are difficult to interpret. In Fig 3b-c, it appears that treatment with CytoD leads to a reduction in G' , and the FWHM data written on Fig. 3e supports this observation. However, the violin plot shows a clear increase in G' with CytoD treatment. What explains this discrepancy?

The CytoD-treated cell cultures resulted in a slight, albeit statistically significant, increase in G' of the background ECM further away from the cell body ($r > 30 \mu\text{m}$), but an obvious decrease in G' in pericellular regions closer to the cell body when compared to the normal condition. Hence, the higher median G' for the CytoD condition on the violin plot in Fig. 3e due to the slight stiffening in the background, but also a smaller FWHM distribution due to the markedly reduced G' closer to the cell (as apparent in Fig. 3a-c). The slight difference in the background G' is not immediately apparent from the color rendering in Fig. 3b-c, especially with a number of pericellular locations having much higher G' than the background. However, this is shown in the figure below, comparing only the background regions away from the cell body. **We stated this observation in lines 226-228 and included the figure below in Supplementary Data 6.**

Figure R1: Comparing backgrounds in two different type of cell cultures in Fig. 3b and 3c. Background elasticity or stiffness is defined as the measured G' in regions that is $> 30 \mu\text{m}$ away from cell body. The statistical significance is indicated as **** p -value < 0.0001 . The median values of stiffness are 107.9 ± 6.3 Pa and 136.5 ± 5.6 Pa for normal and CytoD conditions, respectively.

Additionally, the CytoD treatment is compared to a 'Normal' condition – is 'Normal' a DMSO load control? The reference to the supplemental data suggests that there is not DMSO in these samples, but that should be clarified in the caption.

We thank the referee for indicating this potential point of confusion. Normal refers to the cell culture maintained under general conditions using Dulbecco's Modified Eagle Medium (DMEM)

supplemented with 10% Fetal Bovine Serum and 1% Penicillin-Streptomycin. In contrast, DMSO controls have 50 μL of DMSO added to the sample in the same manner that CytoD treatment was added. This control is necessary due to DMSO being drug vehicle or solvent necessary to dissolve CytoD. The main purpose for a DMSO in control group is to demonstrate the observations in CytoD treated cell cultures resulted from the function of CytoD inhibiting cellular contractility, rather than the addition of the solvent itself. **We have added this clarification to the caption of Fig. 3.**

- The observed decay in elasticity as a function of distance from the cell surface (Fig. 3f, Fig. S10, etc) is attributed in the text to cellular remodeling of the pericellular ECM. However, it is expected that the presence of the cell would impact these measurements. For example, one would expect a similar decay in elasticity if a stiff sphere was embedded within the matrix, regardless of matrix remodeling. Beads near the sphere-ECM interface would displace by a lesser degree as the sphere stiffness increased, even if the surrounding ECM remained at constant G' and G'' . How do the changing mechanics of the cell, in response to CytoD or more broadly as cells spread and change shape, affect measurements in the matrix independent of changes to the matrix due to matrix deposition or proteolysis?

A previous OT-AMR study investigated the presence of a rigid body (20- μm polystyrene beads representing “inactive” cells) in affecting local micromechanical properties of collagen matrices (Ref. 32). Keating, M. *et al* found that embedding 20- μm beads did not affect microscale G' and G'' measured by OT-AMR in comparison to a blank sample without particles. This is likely because such optical manipulation by OT-AMR generates very miniscule degree of local perturbation in the collagen matrix. This perturbation is even smaller with LS-pfOCE due to low-NA radiation pressure forces being much smaller than typical trapping forces generated in OT-AMR. **We have made this note in line 222.**

Regardless, the referee brings up a very interesting question of the effects of changing cell mechanics and morphology on ECM viscoelasticity. We believe it will be extremely challenging to decouple the effect of cell properties and shape independent of cellular activities (*e.g.*, contractility, matrix deposition, and proteolysis), since cell mechanics/morphology and their activities would reciprocally affect each other. We believe it could be ground-breaking to unravel such multi-faceted complexity, but this would require close collaboration with cell biologists/biophysicists/biomaterials scientists in order to independently modulate different aspects of cell mechanics, morphology, and activity while using our technology to characterize ECM mechanics in each condition.

REVIEWERS' COMMENTS

Reviewer #1 (Remarks to the Author):

The authors have submitted a revised manuscript that largely addresses the issues raised by reviewers 2 and 3. Further the authors have addressed my comments on network heterogeneity of fiber network, which are intriguing.

However, I maintain that the impact of this manuscript remains borderline for Nature Communications without a more exciting proof of concept that would be nontrivial to implement using existing techniques. As a methods paper aimed towards the optics community, I think the validation based on "standard control experiments" might be sufficient. However, I am less convinced that this paper will generate as much interest from a mechanobiology or soft materials audience without some new fundamental insights that cannot be revealed any other way.

The authors write in the last sentence of the that: "We believe it could be ground-breaking to unravel such multi-faceted complexity, but this would require close collaboration with cell biologists/ biophysicists/biomaterials scientists in order to independently modulate different aspects of cell mechanics, morphology, and activity while using our technology to characterize ECM mechanics in each condition."

I fully agree with this statement, and I would be much more enthusiastic if these groundbreaking experiments were addressed here instead of relegated to future work.

Minor Comments –

Figure 2

When referring to figure 2 in main text, make clear that ROI1 is in C3 and ROI2 is in C2.

I also suggest that the colorbar for R should be inverted so that red corresponds to solid-like behavior and blue corresponds to fluid-like behavior. Otherwise, the colors scale in the opposite direction from G' and are counterintuitive

Reviewer #2 (Remarks to the Author):

The authors have addressed majority of the questions from the referees. I still find the explanation of G' and G'' slightly vague and not adding to the novelty. The statements of this might be... are not convincing, but on the other hand all other questions and comments were addressed sufficiently. I therefore find the paper adequate for publishing.

Reviewer #3 (Remarks to the Author):

The authors have thoroughly addressed the comments by all reviewers, and the manuscript is suitable for publication.

We appreciate all reviewers' comments during the peer-review process. Your comments inspired us with indispensable suggestions and helped us greatly improve our manuscript. Our point-by-point responses to the referees are discussed below, with changes to the manuscript highlighted in red text.

Reviewer #1 (Remarks to the Author):

The authors have submitted a revised manuscript that largely addresses the issues raised by reviewers 2 and 3. Further the authors have addressed my comments on network heterogeneity of fiber network, which are intriguing.

However, I maintain that the impact of this manuscript remains borderline for Nature Communications without a more exciting proof of concept that would be nontrivial to implement using existing techniques. As a methods paper aimed towards the optics community, I think the validation based on "standard control experiments" might be sufficient. However, I am less convinced that this paper will generate as much interest from a mechanobiology or soft materials audience without some new fundamental insights that cannot be revealed any other way.

The authors write in the last sentence of the that: "We believe it could be ground-breaking to unravel such multi-faceted complexity, but this would require close collaboration with cell biologists/ biophysicists/biomaterials scientists in order to independently modulate different aspects of cell mechanics, morphology, and activity while using our technology to characterize ECM mechanics in each condition."

I fully agree with this statement, and I would be much more enthusiastic if these groundbreaking experiments were addressed here instead of relegated to future work.

We thank the reviewer for their comments. We believe that by addressing the reviewer's comment on network heterogeneity, and by more clearly highlighting that LS-pfOCE has provided access to information that has never been reported by existing techniques (in particular that both softening and stiffening of the ECM occurs after contractility inhibition), our current results will be of sufficient interest to a mechanobiology or soft materials audience.

Minor Comments –

Figure 2

When referring to figure 2 in main text, make clear that ROI1 is in C3 and ROI2 is in C2.

To address this, we have added a more specific indication when discussing about C1 and C2 at Line 188 and Line 190, respectively, in the main text.

I also suggest that the colorbar for R should be inverted so that red corresponds to solid-like behavior and blue corresponds to fluid-like behavior. Otherwise, the colors scale in the opposite direction from G' and are counterintuitive.

Thank you for this helpful suggestion. In response, we have modified Fig. 2 and inverted the color map of relative viscosity R in order to be consistent with the color map of G' .

Reviewer #2 (Remarks to the Author):

The authors have addressed majority of the questions from the referees. I still find the explanation of G prim and G double prim slightly vague and not adding to the novelty. The statements of this might be... are not convincing, but on the other hand all other questions and comments were addressed sufficiently. I therefore find the paper adequate for publishing.

We greatly appreciate the reviewer's comments in helping us revise the manuscript.

Reviewer #3 (Remarks to the Author):

The authors have thoroughly addressed the comments by all reviewers, and the manuscript is suitable for publication.

We express our sincere gratitude to the reviewer for the inspiring comments.